# INTEGER-CENTRIC NEURAL VIDEO COMPRESSION

## ABSTRACT

Cross-platform coding consistency is a fundamental prerequisite for neural video codecs (NVCs). Previous works address this by adopting a **floating-point-centric** perspective to quantize a pretrained floating-point NVC into an integer one. However, this often leads to suboptimal performance with a significant bitrate increase. In this paper, we propose a high-performance cross-platform NVC by designing a comprehensive **integer-centric** training pipeline for model integerization, which enables the training of an integer NVC from scratch. This approach avoids initialization from a floating-point model and allows for more flexible learning across the entire integer space. Observing that the division operations in previous integerization methods destabilize from-scratch training, we propose a multiply-twice integerization strategy to circumvent this instability. Furthermore, we introduce a memorized temporal modeling mechanism, leveraging a memory module to capture long-term dependencies and enhance model capacity. With these innovations, we implement in-loop decoding modules in integer to ensure cross-platform coding consistency, which is further validated across multiple platforms. As a result, our cross-platform NVC achieves an average 20% bitrate reduction compared to H.266/VTM while maintaining an encoding/decoding speed of 153.0/137.3 fps for 1080p video. The code will be released.

## 1 INTRODUCTION

The digital age is generating an unprecedented volume of data, with video content now comprising more than $80\%$ of global traffic Video Marketing Statistics 2025, underscoring the critical importance of video compression technologies. While traditional codecs like H.264/AVC Wiegand et al. (2003), H.265/HEVC HM, H.266/VVC VTM remain widely used, neural video codecs Lu et al. (2019); Liu et al. (2020); Agustsson et al. (2020); Li et al. (2024); Shi et al. (2022); Ho et al. (2022); Kim et al. (2024) have emerged as a promising alternative. Recent work Jia et al. (2025) accelerates high-performance NVCs to achieve real-time coding, pushing NVCs closer to practical deployment.

Despite these achievements, ensuring **cross-platform coding consistency** still remains a key challenge in NVCs. Most NVCs use floating-point arithmetic, which introduces platform-dependent numerical round-off errors. When a video bitstream is distributed and decoded across different platforms, these small errors become non-negligible during entropy decoding. It results in severe distortions in the decoded video, as shown in Figure 1.

In traditional codecs, the cross-platform discrepancies are addressed by using deterministic integer arithmetic in the decoding process. Following their spirits, recent works Le et al. (2022); van Rozendaal et al. (2024); Jia et al. (2025) explore integer-based NVCs for cross-platform coding consistency. However, these methods come at the cost of increased computational burden or significant performance degradation. For instance, the 16-bit integerization in DCVC-RT Jia et al. (2025) slows coding by $5\times$, while 8-bit quantization in MobileNVC van Rozendaal et al. (2024) causes an 80% BD-Rate penalty. Currently, developing a cross-platform NVC with both real-time coding speed and high compression efficiency remains an open problem.

In this paper, we present a new perspective on integerization for cross-platform NVCs. Existing approaches Le et al. (2022); van Rozendaal et al. (2024); Jia et al. (2025) are **floating-point-centric**, *i.e.*, developing the floating-point model first and then finetuning to an integer version. The quantization focuses on preserving the performance of the original model, typically by using a much smaller learning rate (*e.g.*, $0.01\times$ in MobileNVC) for quantization-aware training. However, this

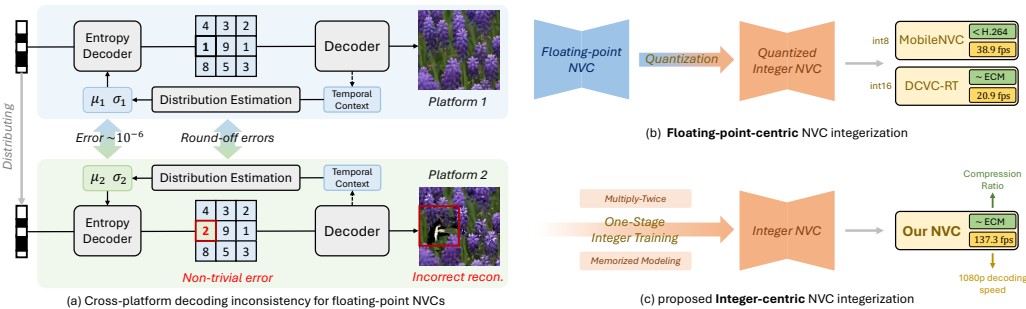

Figure 1: Cross-platform neural video compression. (a) Round-off errors in floating-point NVCs can lead to severe reconstruction distortions across different platforms. (b) Existing floating-point-centric quantized integer NVCs fail to achieve both a satisfactory compression ratio and real-time coding speed. (c) Adopting an integer-centric approach, our NVC ensures cross-platform coding consistency, real-time coding and high compression efficiency.

limits the optimization space for integer model parameters, often resulting in sub-optimal compression performance. Moreover, the architectural design of integer and floating-point models differs, *e.g.*, integer models favor efficient activations like ReLU Agarap (2018) instead of complex ones like GeLU Hendrycks & Gimpel (2016). Such architectural differences are difficult to accommodate through direct quantization. These challenges raise a natural question: instead of quantizing from a floating-point model, can we design an integer model from scratch?

We adopt this **integer-centric** perspective to develop a high-performance cross-platform NVC. With careful studies and designs, our NVC can now be trained from scratch as an integer model rather than being quantized from a floating-point model, granting model parameters the flexibility to be learned across the entire integer space. In addition, we observe that conventional quantization involves multiplication and division by the quantization step, which severely destabilizes from-scratch integer training and ultimately leads to model collapse. We address this by introducing a *multiply-twice* strategy that avoids divisions, resulting in a stable training process. To enhance model capacity, we propose a memorized temporal modeling mechanism that incorporates a dynamically updated memory feature to capture long-term temporal dependencies. It effectively expands the capacity and improves compression performance, especially for low-precision integer NVCs. With these innovations, we implement in-loop decoding modules in integer for cross-platform coding consistency.

Our integer-centric NVC guarantees cross-platform coding, which is empirically validated across multiple platforms, including NVIDIA GPUs and Intel integrated graphics. It achieves an average 20% bitrate reduction over H.266/VTM, significantly outperforming existing 8-bit integer NVCs like MobileNVC. In terms of coding speed, our NVC reaches 153.0/137.3 FPS for 1080p video on an NVIDIA A100 GPU, surpassing real-time NVC, DCVC-RT.

We summarize the contributions of this paper as follows:

- A comprehensive integer-centric training pipeline to enable training an integer NVC from scratch.
- A novel integerization strategy that stabilizes the training and improves the compression ratio.
- An effective temporal modeling mechanism that equips NVC with memorization capabilities.
- Based on these designs, we develop a high-performance cross-platform NVC that achieves real-time 1080p coding on consumer hardware with a 20% bitrate reduction compared to VTM/H.266.

## 2 RELATED WORKS

### 2.1 PRACTICAL NEURAL VIDEO COMPRESSION

Recent advances Lu et al. (2019; 2020b); Yang et al. (2020); Agustsson et al. (2020); Shi et al. (2022); Liu et al. (2023a); Ho et al. (2022); Lu et al. (2024); Qi et al. (2024); Van Rozendaal et al. (2021); Guo et al. (2023) significantly improved the rate-distortion performance of Neural Video Codecs (NVCs), enabling them to surpass traditional codecs Li et al. (2024); Sheng et al. (2024) in compression

ratio. With compression efficiency largely solved, research has shifted toward making NVCs more practical and functional, including rate control Zhang et al. (2023); Rippel et al. (2021); Li et al. (2024) and random access Chen et al. (2023) supports. To enable real-time coding, recent efforts investigate INR-based low-complexity decoding Kim et al. (2024); Hu & Xu (2023); Gao et al. (2024), efficient network designs Le et al. (2022); van Rozendaal et al. (2024), or reducing the operational complexity Jia et al. (2025) of NVCs. While these advancements have significantly pushed NVCs toward practical implementation, a critical prerequisite for deployment, *i.e.*, cross-platform coding consistency, remains largely underexplored.

## 2.2 CROSS-PLATFORM CODING CONSISTENCY

Across different platforms, platform-dependent round-off errors can lead to significant discrepancies in entropy decoding, resulting in incorrect reconstructions. Tian et al. (2023) propose a calibration bitstream to correct these errors, but it fails when errors accumulate beyond the predefined precision threshold in a coding chain. A more robust solution lies in deploying integer-based NVCs, which leverage deterministic integer arithmetic to inherently resolve cross-platform coding consistency issues. While DCVC-RT Jia et al. (2025) adopts a 16-bit integer model, it suffers from a 5× slowdown in coding speed due to limited hardware optimization for int16 inference. On the other hand, MobileNVC van Rozendaal et al. (2024) uses 8-bit quantization for speed but sacrifices compression quality with an 80% bitrate increase due to reduced precision. This work aims to build an 8-bit integer NVC that balances high compression performance and low latency, overcoming previous limitations.

## 2.3 MODEL QUANTIZATION

Model quantization converts floating-point models into low-precision integer models. While post-training quantization Liu et al. (2021); Nahshan et al. (2021) and quantization-aware training Zhu et al. (2016); Polino et al. (2018); Esser et al. (2020); Lu et al. (2023) often maintain accuracy in general tasks, they perform poorly in video compression due to sensitivity in entropy modeling. For example, MobileNVC van Rozendaal et al. (2024) adapts existing quantization techniques to neural video compression models but shows a 80% performance drop on HEVC Class B.

To address this, we propose training an integer NVC from scratch to preserve RD performance while ensuring quantization efficiency and stability. Notably, Wang et al. Wang et al. (2023) also train binary large language models (LLMs) from scratch. However, their rely on a custom BitLinear layer to replace standard linear layers, making it hard to extend to other tasks. In contrast, our integer-centric training pipeline is architecture-agnostic and can be easily generalized to other structures.

# 3 INTEGER-CENTRIC MODEL INTEGERIZATION

## 3.1 PRELIMINARY

A common approach to ensuring cross-platform coding consistency is model integerization. By quantizing model weights and features into integers, deterministic computations are enforced to guarantee consistent outputs across different devices. Despite its importance, only a few works have explored integerization in neural video codecs. MobileCodec Le et al. (2022) was the first to introduce quantization-aware training Nagel et al. (2021) for an efficient 8-bit integer neural video codec. Building on this, MobileNVC van Rozendaal et al. (2024) leverages the more advanced learned step size quantization (LSQ Esser et al. (2020)) to further improve compression efficiency.

Here, we briefly introduce how LSQ quantize model weights and latent features into integers. Given an input value $v$ and a learned quantization step size $s$, the quantization process is defined as:

$$\bar{v} = \lfloor \text{clip}(v/s, -Q_N, Q_P) \rceil, \quad \hat{v} = \bar{v} \times s \tag{1}$$

When quantizing into $b$ bits, $Q_P = 2^{b-1} - 1$ and $Q_N = 2^{b-1}$ represent the number of positive and negative quantization levels, respectively. Here, $\bar{v}$ is is the quantized integer-scaled representation $v$,

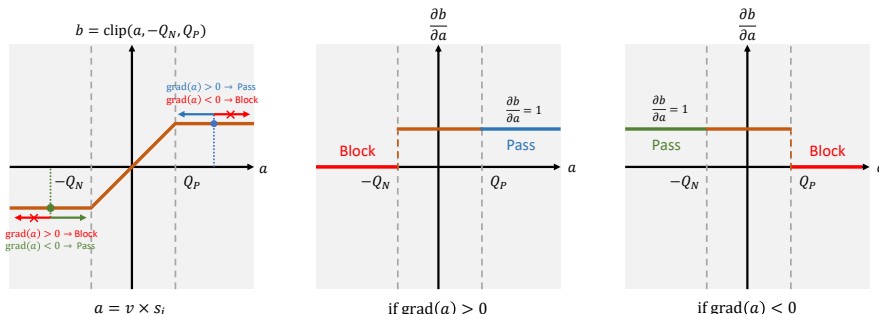

Figure 2: Clipping gradient pass. When computing the out-of-boundary clipping gradient, we allow the gradient to pass through if its direction pulls the feature back within the boundary.

and $\hat{v}$ is the quantized representation at the same scale as $v$. In training, the gradients are computed:

$$\frac{\partial \hat{v}}{\partial s} = \begin{cases} -v/s + \lfloor v/s \rceil, & \text{if } -Q_N < v/s < Q_P, \\ -Q_N, & \text{if } v/s \le -Q_N, \\ Q_P, & \text{if } v/s \ge Q_P. \end{cases} \tag{2}$$

$$\frac{\partial \hat{v}}{\partial v} = \begin{cases} 1, & \text{if } -Q_N < v/s < Q_P, \\ 0, & \text{otherwise.} \end{cases} \tag{3}$$

Existing integer NVCs follow a **floating-point-centric** approach, *e.g.*, MobileNVC first trains a floating-point codec and then quantizes it into an 8-bit using LSQ. While this enables cross-platform consistency, it leads to a significant drop in compression ratio (as discussed in Section 2.3).

This performance gap raises a critical question: Is quantizing a floating-point NVC truly the best path to building an integer NVC? Or can we instead design an integer NVC from the ground up, tailored specifically to the characteristics of integer models and unconstrained by floating-point design? Motivated by this **integer-centric** perspective, we revisit the core components of cross-platform design to bridge the performance gap and build a high-efficiency cross-platform NVC.

### 3.2 ONE-STAGE INTEGER TRAINING

Floating-point-centric methods quantize floating-point models into integer using techniques like quantization-aware training (QAT). However, our analysis reveals two key limitations:

- First, quantization implicitly assumes that a low-precision integer codec should retain the computational behavior of its floating-point counterpart. For example, MobileNVC applies QAT with a learning rate $0.01\times$ smaller than in floating-point training to maintain the computational behavior. However, forcing a low-precision model to fit a floating-point feature space may not lead to the most effective solution for integer computation.

- Second, the optimal architecture for an integer model may differ from a floating-point model. For instance, integer NVCs favor simple activations like ReLU Agarap (2018) for efficient implementation, whereas floating-point NVCs explore more complex activations like GeLU Hendrycks & Gimpel (2016) to enhance representational power. This suggests that quantizing a floating-point model may not fully exploit the potential of an integer model design.

To overcome these limitations, a natural alternative is to **train an integer model from scratch**. This one-stage integer training approach allows direct training on a model structure designed for integer calculations, eliminating the constraints posed by a floating-point initialization.

### 3.3 MULTIPLY-TWICE INTEGERIZATION

In the LSQ integerization process (Equation 1), a value $v$ is first divided by the step size $s$ to convert to integer scale $\bar{v}$, and then multiplied by $s$ to recover the original scale $\hat{v}$. This ensures $\hat{v}$ maintains the same scale as $v$, preserving floating-point computational behavior. However, the division can

cause unexpected feature and gradient values and make training unstable when $s$ is small. This issue becomes particularly severe when applying cascaded training Lu et al. (2020a); Sheng et al. (2022); Li et al. (2024) to boost NVC performance, where gradients accumulate over long frame sequences.

**Multiply-twice with decoupled step sizes**. In one-stage integer training, the quantized $\hat{v}$ does not need to maintain the same scale as the original $v$, since alignment with a floating-point model is unnecessary. The model weights can naturally adapt to different scales to enable exploration in the entire integer space. Based on this, we propose to decouple the step size $s$ into two separate factors: $s_i$ for converting $v$ into integer scale $\bar{v}$ and $s_c$ for mapping $\bar{v}$ into a calculation-aligned[1] scale $\hat{v}$. This avoids division and uses two multiplications instead to significantly enhance training stability.

$$\hat{v} = \lfloor \text{clip}(v \times s_i, -Q_N, Q_P) \rceil \times s_c = \begin{cases} \lfloor v \times s_i \rceil \times s_c, & \text{if } -Q_N < v \times s_i < Q_P, \\ -Q_N \times s_c, & \text{if } v \times s_i \leq -Q_N, \\ Q_P \times s_c, & \text{if } v \times s_i \geq Q_P. \end{cases} \quad (4)$$

To compute the gradient, we employ the straight-through estimator (STE) Bengio et al. (2013) to approximate the gradient of the rounding operation, allowing the gradients to be computed as:

$$\frac{\partial \hat{v}}{\partial s_i} = \begin{cases} v \times s_c, & \text{if } -Q_N < v \times s_i < Q_P, \\ 0, & \text{otherwise.} \end{cases} \quad (5)$$

$$\frac{\partial \hat{v}}{\partial s_c} = \begin{cases} \lfloor v \times s_i \rceil, & \text{if } -Q_N < v \times s_i < Q_P, \\ -Q_N, & \text{if } v \times s_i \leq -Q_N, \\ Q_P, & \text{if } v \times s_i \geq Q_P. \end{cases} \quad (6)$$

$$\frac{\partial \hat{v}}{\partial v} = \begin{cases} s_i \times s_c, & \text{if } -Q_N < v \times s_i < Q_P, \\ 0, & \text{otherwise.} \end{cases} \quad (7)$$

**Clipping gradient pass**. In Equation 5, when $v \times s_i$ exceeds value bound, the gradient of $s_i$ becomes 0 due to the clipping operation, preventing its update. However, ideally, we want to update $v \times s_i$ to bring it back within the range. To achieve it, we follow prior works[2] to introduce a gradient pass mechanism that allows gradients to propagate to $s_i$ when out of bounds. Specifically, if $v \times s_i > Q_P$ and he gradient direction indicates it should decrease, *i.e.*, $\text{grad}(v \times s_i) > 0$, we allow the gradient to pass to $s_i$. The same goes for $v \times s_i < -Q_N$ and $\text{grad}(v \times s_i) < 0$. This can be formulated by:

$$\frac{\partial \hat{v}}{\partial s_i} = \begin{cases} v \times s_c, & \text{if } c_{in} \text{ or } c_{up} \text{ or } c_{low}, \\ 0, & \text{otherwise.} \end{cases} \quad (8)$$

where $c_{in} = -Q_N < v \times s_i < Q_P$, $c_{up} = (v \times s_i \geq Q_P) \wedge (\text{grad}(v \times s_i) > 0)$ and $c_{low} = (v \times s_i \leq -Q_N) \wedge (\text{grad}(v \times s_i) < 0)$. Notably, the clipping gradient pass works well with our decoupled step sizes, as the gradients of $s_i$ and $s_c$ are independently. If it is applied in LSQ, the coupled gradient may cause an incorrect updating of $s$. In our NVC, removing this mechanism causes a 14% performance drop, highlighting its importance in the multiply-twice integerization process.

## 4 CROSS-PLATFORM NEURAL VIDEO COMPRESSION

### 4.1 OVERVIEW

In this section, we present the details of the proposed high-performance cross-platform neural video codec. Our model design builds upon DCVC-RT Jia et al. (2025). The framework of the proposed NVC is illustrated in Figure 3. For each input frame $I_t$, we first patchfy it into $\frac{1}{8}$-scale features using patch embedding Dosovitskiy et al. (2021). Next, a feature extractor retrieves temporal context from a memory feature (Section 4.2), which stores long-term temporal information. This extracted temporal context is then used for conditional coding Li et al. (2021) to compress the $\frac{1}{8}$-scale features. The decoded features are subsequently merged into the memory model to update the memory feature for the next coding loop. Finally, to reconstruct the frame, the decoded features are processed through a reconstruction generation module, producing the reconstructed frame $\hat{I}_t$.

---

[1] In integer calculations, each integer value is at a different scale determined by $s_i$. Therefore, before passing the computed results to the next calculation, they must be converted to an aligned scale using $s_c$ to neutralize the effect of the previous scale conversion.

[2] Gaussian scale clipping gradient in `https://github.com/InterDigitalInc/CompressAI`.

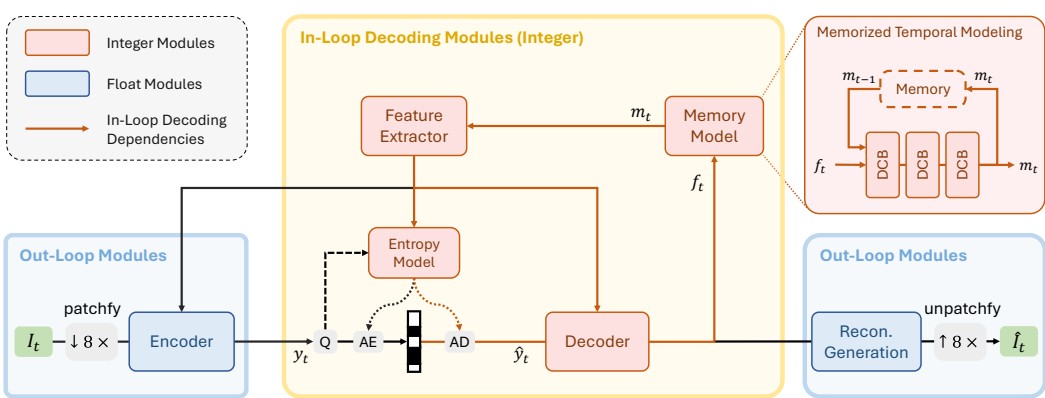

Figure 3: Framework Overview. DCB, Q, AE, AD denote the depth-wise convolution block, quantization, arithmetic encoder, arithmetic decoder, respectively. In our NVC, the in-loop decoding modules are implemented in integer to maintain entropy coding consistency across platforms, while the out-loop modules remain floating-point to improve performance.

## 4.2 MEMORIZED TEMPORAL MODELING

Model capacity is critical for NVCs to enable effective feature transformation and better rate-distortion performance. However, the low precision of integer calculations can hinder the codec to learn sufficient prior knowledge from large-scale data, ultimately limiting its model capacity. It makes model capacity even more crucial for integer NVCs. Recent advancements in large language models and intelligent agents Zhong et al. (2024); Liang et al. (2023); Liu et al. (2023b) suggest that model capacity depends not only on model parameters but also on the ability to represent and retain useful historical context. Motivated by this, we propose an effective memorized temporal modeling module that stores and updates temporal context, granting the NVC model with memorization capabilities.

The previous state-of-the-art NVC model Jia et al. (2025) utilizes only a simple buffer that stores a single previously decoded feature for temporal context extraction. In contrast, we introduce memorized temporal modeling, where a dynamically updated memory feature learns and retains long-term historical information. As illustrated in Figure 3, we introduce a memory feature $m_t$ to store temporal history information from the decoded feature $f_t$ and assist in temporal context extraction. After decoding $f_t$, the previous memory feature $m_{t-1}$ is updated into $m_t$. This update is performed by concatenating $f_t$ and $m_{t-1}$ along the channel dimension, followed by processing through cascaded depth-wise convolution blocks. The memory size of $m_t$ remains constant across different timesteps to maintain efficiency. This also allows the memory module to selectively retain the most relevant context while discarding less important temporal information. Through experiments, we observe that constructing a more representative memory feature significantly improves the compression ratio in a floating-point model, with an even greater performance gain in our integer NVC.

## 4.3 IN-LOOP MODEL INTEGERIZATION

As discussed in Section 3, the cross-platform coding consistency issue primarily arises due to round-off errors in entropy decoding. To ensure consistency, all modules involved in entropy coding must follow deterministic integer calculations. In our NVC, we categorize the modules into two classes:

- In-loop decoding modules, including the decoder, feature extractor, memory updater and entropy model. These modules operate during decoding, and their outputs influence the decoding process of subsequent frames (termed as *in-loop*). To ensure consistent entropy coding, we implement these modules using integer arithmetic.

- Out-loop modules, including the encoder, hyperprior encoder and reconstruction generation module. These modules are outside the decoding loop and do not affect decoding consistency, so we retain them in floating-point to improve compression performance.

Notably, traditional coding standards typically define the decoding process while leaving the encoding process open, and our NVC framework goes a step further by opening the reconstruction generation module. Positioned outside the decoding loop, it can operate in floating-point, resulting in a 3.4% improvement in compression ratio, as shown in Table 1.

### 4.4 Integer Module Implementation

In this section, we briefly introduce the implementation of our integer modules. Due to the complexity of algorithm design and engineering in integer implementation, we focus on the high-level concepts here, with a detailed algorithm description provided in the supplementary material.

**QP-aware quantization step sizes**. Following Jia et al. (2025), we enable variable bitrates within a single model. Given the significant variation in feature distribution across different bitrates, we learn distinct step sizes $s_i$ and $s_c$ for each quantization parameter (QP). During inference, the step sizes are selected based on the given QP. Experimental results demonstrate that QP-aware step sizes bring approximately 15% bitrate savings compared to uniform step sizes across different QPs.

**Inference pipeline**. In a convolution, the features and weights are first converted to int8 using:

$$v^{\text{int8}} = \left\lfloor \text{clip}(v^{\text{float}} \times s_i, -Q_N, Q_P) \right\rceil \tag{9}$$

The convolution is then carried out entirely in the integer domain. Since the output values remain in a scale $s_i$ specific to the current convolution layer, they are converted to a calculation-aligned scale using a step size $s_c$ before further calculations (*e.g.*, adding a bias).

**Overflow prevention**. Different platforms may exhibit varying behaviors when integer calculations overflow, making overflow prevention crucial for ensuring cross-platform coding consistency. To address this, we set the accumulator data type to int32 during convolution and split the accumulated result into its higher 16-bit and lower 16-bit parts to avoid overflow. Our implementation can prevent overflow under **any** circumstances in our NVC, with further details provided in the supplementary.

**Cross-platform consistency verification**. We implement our NVC on multiple platforms to verify coding consistency. As shown in Table 2, our NVC produces precisely consistent coding results across platforms, matching the single-device output exactly. Currently, we support NVIDIA GPUs (A100, RTX 4090, RTX 2080Ti) and Intel integrated graphics (Arc™ 140V). CPU support is not included due to the significant engineering effort required for custom operations like integer convolution. As an academic prototype, we leave CPU implementation for future work after code release.

Table 1: Comparison on decoding integerization.

| Integerization | BD-Rate |
|---|---|
| In-Loop Decoding | 0% |
| Full Decoding | 3.4% |

Table 2: Cross-platform coding consistency verification.

| Settings | Single | NVIDIA GPU → NVIDIA GPU | | | | | | NVIDIA GPU → Intel Integrated Graphics | |
|---|---|---|---|---|---|---|---|---|---|
| Enc. Platform | A100 | A100 | A100 | 4090 | 4090 | 2080Ti | 2080Ti | 2080Ti | Intel® Arc™ 140V |
| Dec. Platform | A100 | 4090 | 2080Ti | A100 | 2080Ti | A100 | 4090 | Intel® Arc™ 140V | 2080Ti |
| BD-Rate | 0% | 0% | 0% | 0% | 0% | 0% | 0% | 0% | 0% |

\* Currently, CPU is not tested due to heavy engineering effort for implementing integer operations on CPU.

## 5 Experiments

### 5.1 Settings

**Datasets.** We train our NVC on the Vimeo-90k dataset Xue et al. (2019) using 7-frame sequences. For fine-tuning, we follow Li et al. (2024) by processing the original Vimeo videos ori to create extended sequences. Evaluation is conducted on the HEVC Class B–E Flynn et al., UVG Mercat et al. (2020), and MCL-JCV Wang et al. (2016) datasets.

**Evaluation Details.** For traditional codecs, we compare with HM HM (the best H.265 encoder), VTM VTM (the best H.266 encoder), and ECM ECM (a prototype of the next-generation traditional codec). Configuration details are in the supplementary material. For neural codecs, we benchmark state-of-the-art methods, including DCVC-DC Li et al. (2023), DCVC-FM Li et al. (2024), and DCVC-RT Jia et al. (2025). Rate-distortion performance is measured using BD-Rate Bjontegaard (2001), and coding speed is evaluated across various QP values on $1920 \times 1080$ resolution using a single NVIDIA A100 GPU and an AMD EPYC 7V13 processor. Following the protocol in Jia et al.

Table 3: BD-Rate (%) comparison in YUV420 colorspace. All frames with intra-period=−1.

| Method | UVG | MCL-JCV | HEVC B | HEVC C | HEVC D | HEVC E | Average | Coding Speed Enc. | Coding Speed Dec. |
|---|---|---|---|---|---|---|---|---|---|
| *Traditional Codecs* | | | | | | | | | |
| VTM-17.0 | 0.0 | 0.0 | 0.0 | 0.0 | 0.0 | 0.0 | 0.0 | 0.01 fps | 23.6 fps |
| HM-16.25 | 40.1 | 48.6 | 47.6 | 41.0 | 34.5 | 42.8 | 42.4 | 0.05 fps | 39.6 fps |
| ECM-11.0 | −20.0 | −22.1 | −22.2 | −21.2 | −20.4 | −17.2 | −20.5 | 0.002 fps | 3.4 fps |
| *Single-Platform NVCs* | | | | | | | | | |
| DCVC-DC | 6.5 | −4.4 | 13.1 | −3.4 | −14.8 | 90.2 | 14.5 | 3.3 fps | 4.3 fps |
| DCVC-FM (fp16) | −16.8 | −8.0 | −15.4 | −30.2 | −37.5 | −20.2 | −21.3 | 5.0 fps | 5.9 fps |
| DCVC-RT (fp16) | −24.0 | −14.8 | −16.6 | −21.0 | −27.3 | −22.4 | −21.0 | 125.2 fps | 112.8 fps |
| *Cross-Platform NVCs* | | | | | | | | | |
| DCVC-RT (int16) | −21.0 | −12.3 | −14.8 | −20.0 | −26.4 | −15.0 | −18.3 | 28.3 fps | 20.9 fps |
| **Our NVC** | −22.7 | −9.4 | −14.0 | −20.2 | −26.5 | −27.0 | −20.0 | 153.0 fps | 137.3 fps |

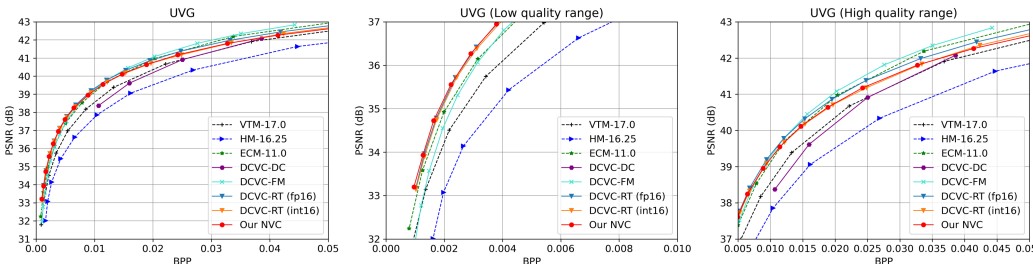

Figure 4: Rate-distortion curves on UVG dataset. All frames are tested in YUV420 colorspace with intra-period=−1. Results on more datasets are provided in the supplementary materials.

(2025), we test all frames with an intra-period of −1, on both YUV420 and RGB colorspaces under low-delay conditions, using actual encoded bitstreams for bitrate calculation.

**Training Details.** To support variable-rate compression within a single model, we randomly assign QP values in the range $[0, 63]$ during training. Following Li et al. (2024), we apply a hierarchical weight setting to the distortion term, interpolating $\lambda$ values between 1 and 768. Moreover, we use a combined distortion loss in both YUV and RGB colorspaces to support them within a single model.

## 5.2 COMPARISON WITH SOTA METHODS

We present the BD-Rate and coding speed comparisons in Table 3. Using VTM as the anchor, our NVC achieves an average BD-Rate of −20.0%, comparable to the best traditional codec, ECM (−20.5%), and state-of-the-art single-platform NVCs like DCVC-FM (−21.3%) and DCVC-RT (fp16, −21.0%). Additionally, our NVC outperforms the cross-platform NVC DCVC-RT (int16, −18.3%) in compression performance. In terms of coding speed, our NVC achieves 153.0 fps for encoding and 137.3 fps for decoding, which is the fastest among all methods. Compared to cross-platform DCVC-RT (int16), it is 5.4× and 6.6× faster for encoding and decoding, respectively. This significant speedup is attributed to our efficient implementation of 8-bit integer modules, which are considerably faster than 16-bit integer calculations. Additional results on the RGB colorspace are provided in the supplementary material. Figure 4 presents the rate-distortion curves on the UVG dataset. These curves show that our NVC achieves the best compression performance at low bitrates. At higher bitrates, performance slightly declines due to model capacity limitations imposed by integer precision. However, our NVC remains competitive with DCVC-RT and consistently outperforms VTM across all bitrates.

Table 4: Ablation study.

| ID | Settings | BD-Rate |
|---|---|---|
| | *Float training (single-platform)* | |
| A | DCVC-RT (fp16) $\rightarrow$ Anchor | 0% |
| | *Quantizing a floating-point model* | |
| $B_1$ | A + Learned step size quantization | 52.1% |
| $B_2$ | $B_1$ + Multiply-twice integerization | 56.1% |
| $B_3$ | $B_2$ + In-loop integerization | 45.4% |
| | *One-stage integer training* | |
| $C_1$ | A + Learned step size quantization | >500% |
| $C_2$ | $C_1$ + Multiply-twice integerization | 30.6% |
| $C_3$ | $C_2$ + In-Loop integerization | 15.4% |
| | *Network design* | |
| D | $C_3$ + Memorized temporal modelling $\rightarrow$ **Proposed cross-platform NVC** | 1.9% |

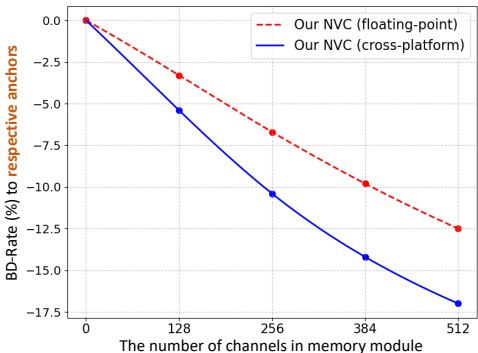

Figure 5: Ablation study on memorized temporal modeling. The floating-point model follows the same network structure as cross-platform NVC but is trained in floating-point for comparison.

## 5.3 ABLATION STUDY

**Add-on Study**. In Table 4, we use DCVC-RT (fp16) as an anchor (ID=A in the Table) and incrementally introduce our proposed design components to analyze their impact on high-performance NVCs. First, we apply standard LSQ to quantize a floating-point NVC into an fully 8-bit integer model following MobileNVC van Rozendaal et al. (2024) ($B_1$), resulting in a severe 52.1% performance drop. Since the training objective is to preserve the performance of the floating-point model, multiply-twice integerization ($B_2$) does not provide additional flexibility, as the model gets trapped in local optima, leading to no performance improvement.

Next, we train an integer NVC from scratch using one-stage integer training. However, LSQ introduces severe instability, causing the model to crash with extremely high loss ($C_1$). In contrast, multiply-twice integerization ($C_2$) effectively stabilizes training, already surpassing LSQ quantization ($B_1$) with a BD-Rate of 30.6%. Further improvement is achieved by keeping only the in-loop decoding modules in integer precision while retaining the rest of the model in floating point ($C_3$), reducing the BD-Rate to 15.4%. Finally, integrating memorized temporal modeling significantly enhances model capacity, further improving performance to 1.9%.

**Memorized temporal modeling**. In Section 4.2, we introduced a memory module to store historical temporal information, thereby enhancing the model capacity of NVCs. Figure 5 investigates how varying the size of the memory module affects the compression ratio. Using the floating-point version of our NVC as a baseline, increasing the memory module size up to 512 yields a gradual bit saving of up to 12.5%, demonstrating that the memory module effectively captures long-term temporal context to improve temporal modeling capability. Notably, when using our cross-platform NVC as the baseline, we observe an even larger performance gain of up to 17.0%. This suggests that low-precision integer NVCs, which are more constrained in capacity, benefit significantly from the enhanced capacity provided by the memory module. In our NVC we adopt 384 channels to balance rate-distortion-complexity trade-off.

## 6 CONCLUSIONS

In this paper, we propose a high-performance cross-platform NVC in an integer-centric perspective, designing the model from the ground up rather than quantizing a floating-point one. We introduce one-stage integer training with a multiply-twice integerization process for more stable, flexible training. To enhance model capacity, we propose a memorized temporal modeling mechanism with a memory module to capture long-term dependencies. To our knowledge, this is the first cross-platform NVC achieving 130 fps 1080p coding with an average 20% bitrate reduction over H.266/VTM, marking a significant step toward practical NVCs.

**Limitations**. Although our NVC achieves fast coding on GPU platforms, it has not been optimized for mobile platforms, where computational resources are constrained. This affects its practical applicability. In the future, we will explore efficient implementations for a practical mobile NVC.

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

# A  TEST SETTINGS

For a fair comparison with traditional codecs, we use their optimal low-delay settings to achieve the best possible compression ratio. We evaluate them in both YUV420 and RGB colorspaces to ensure a comprehensive assessment.

**YUV420 colorspace**. Our primary evaluation focuses on the YUV420 colorspace, which is extensively optimized in traditional video codecs. We compare our approach against HM HM, VTM VTM, and ECM ECM, representing the best H.265 encoder, the best H.266 encoder, and the prototype of the next-generation traditional codec, respectively. For each codec, we use the officially configuration files: *encoder_lowdelay_main10.cfg*, *encoder_lowdelay_vtm.cfg*, and *encoder_lowdelay_ecm.cfg*. The parameters for coding are:

- -c {*config file name*}
  --InputFile={*input video name*}
  --InputBitDepth=8
  --OutputBitDepth=8
  --OutputBitDepthC=8
  --FrameRate={*frame rate*}
  --DecodingRefreshType=2
  --FramesToBeEncoded={*frame number*}
  --SourceWidth={*width*}
  --SourceHeight={*height*}
  --IntraPeriod={*intra period*}
  --QP={*qp*}
  --Level=6.2
  --BitstreamFile={*bitstream file name*}

**RGB colorspace**. Since the raw videos in our experiments are stored in YUV420 format, we convert them to RGB for testing. Following JPEG AI Alshina et al. (2022); Anchors · JPEG-AI MMSP Challenge and Li et al. (2023; 2024); Jia et al. (2025), we use the BT.709 conversion, which achieves a higher compression ratio than the commonly used BT.601. Traditional codecs are tested using 10-bit YUV444 as the internal colorspace, with final evaluations conducted in RGB. Prior studies Li et al. (2023; 2024); Jia et al. (2025) have shown that this approach results in better compression performance than directly encoding in RGB. For HM, VTM, and ECM, we use the following configuration files: we utilize *encoder_lowdelay_rext.cfg*, *encoder_lowdelay_vtm.cfg*, and *encoder_lowdelay_ecm.cfg* as the config file, respectively. The parameters for coding are:

- -c {*config file name*}
  --InputFile={*input file name*}
  --InputBitDepth=10
  --OutputBitDepth=10
  --OutputBitDepthC=10
  --InputChromaFormat=444
  --FrameRate={*frame rate*}
  --DecodingRefreshType=2
  --FramesToBeEncoded={*frame number*}
  --SourceWidth={*width*}
  --SourceHeight={*height*}
  --IntraPeriod={*intra period*}
  --QP={*qp*}
  --Level=6.2
  --BitstreamFile={*bitstream file name*}

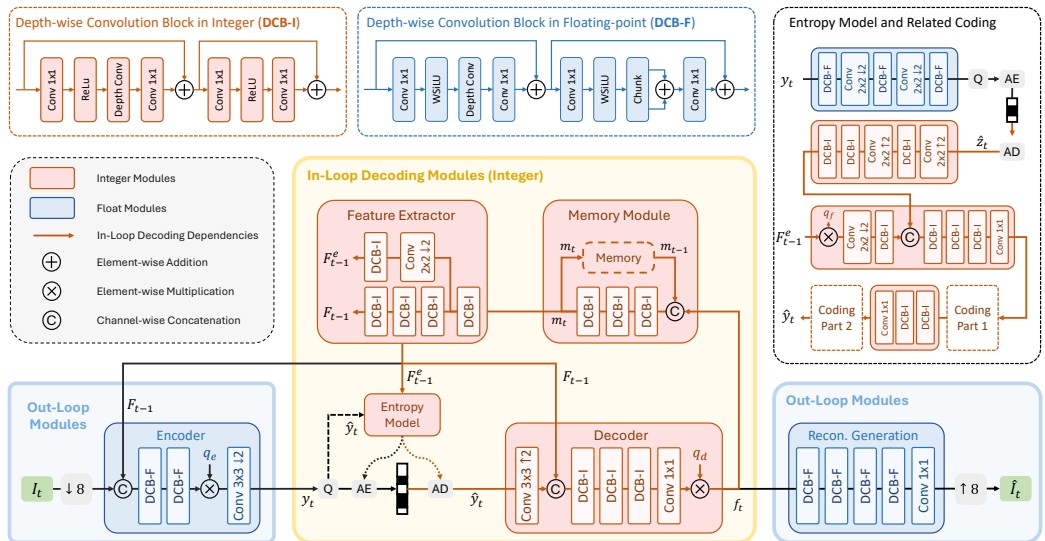

Figure 6: Model Structures. Q, AE, AD denote quantization, arithmetic encoder, decoder, respectively.

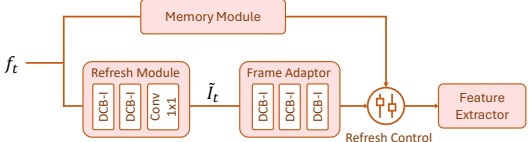

Figure 7: Feature refresh in our NVC.

# B  IMPLEMENTATION DETAILS

## B.1  MODEL STRUCTURES

In this section, we introduce the detailed module structure of our NVC. As shown in Fig. 6, our NVC consists of six modules, categorized into two groups: 1) in-loop decoding modules, including the decoder, memory model, feature extractor, and entropy model (excluding the hyper encoder).; 2) out-loop modules, ncluding the encoder, reconstruction generation module, and the hyper encoder. The in-loop decoding modules are implemented in integer, while the out-loop modules operate in floating-point.

**Building blocks**. We design distinct building blocks (*i.e.*, the depth-wise convolution blocks, DCB) for in-loop and out-loop module. For floating-point modules, the DCB follows the design in Jia et al. (2025) to enhance representation capability. For integer modules, the DCB employs a simple ReLU activation for efficient implementation and removes the *chunk-add* operation to support larger channels.

**Entropy coding**. We split $y$ into two parts and adopt a two-step entropy coding scheme Li et al. (2022). Additionally, we implement parallel coding Jia et al. (2025) to conduct some modules parallel with entropy coding in parallel for acceleration.

**Variable bitrates**. Following Jia et al. (2025), we incorporate a rate-adjustment module bank to learn 1) separate factorized priors for hyper information coding and 2) modulation vectors $q_e$, $q_d$ and $q_f$ in encoder, decoder and entropy model, respectively. This enables support for variable bitrates within a single module and rate-control functionality.

### B.2 Feature Refresh Mechanism

DCVC-FM Li et al. (2024) introduces a periodic feature refresh mechanism to mitigate temporal quality degradation in long prediction chains. This is achieved by periodically extracting temporal context from the decoded frame $\hat{I}_t$ instead of decoded feature $f_t$. However, in our NVC, the reconstruction generation module operates in floating-point. Directly extracting temporal context from $\hat{I}_t$ would introduce nondeterministic calculations during frame decoding after refresh, potentially leading to cross-platform coding consistency issues. To address this, we introduce an additional refresh module that generates a new reconstruction $\tilde{I}_t$ from $f_t$. As shown in Fig. 7, this module consists of two integer DCBs, ensuring that the generation of $\tilde{x}_t$ remains deterministic, thereby maintaining cross-platform coding consistency.

## C INTEGER MODULE IMPLEMENTATION

In our integer modules, there are three kinds of main operations: simple arithmetic (*e.g.*, addition and concatenation), convolutions, and activations (*i.e.*, ReLU). For all operations, input and output features are set to int16. For convolutions which dominate computational cost, int16 features are converted to int8 to improve processing speed. In this section, for clarity, we mark the data type of each value at the top right corner. The hyperparameters $K_{a,\,b,\,c,\,d}$ are set to int32 but are auto-cast to other types when necessary.

**Int16 conversion**. We first introduce how we convert floating-point features into int16 in our NVC. Specifically, each int16 feature $x^{\text{int16}}$ is linearly mapped from its corresponding floating-point value $x^{\text{float}}$ via:

$$x^{\text{int16}} = \left\lfloor \text{clip}(K_a \times x^{\text{float}}, -32768, 32767) \right\rceil .\text{to\_int16}() \tag{10}$$

We empirically set $K_a = 2^{10}$, such that the floating-point range $[-32.0, 31.999]$ is mapped to the int16 range $[-32768, 32767]$. In our implementation, we use extensive kernel fusions to combine different operations into a single calculation unit to optimize GPU memory I/O of these int16 features.

**Simple arithmetic**. Simple arithmetic operations are performed directly on int16 for simplicity. In this sense, arithmetic operations between two features can be performed directly "as if" they were floating-point values, eliminating the hassle of accounting for different factors, $s_i$ and $s_c$, which vary across layers if the features were designed to be int8. We observe the aforementioned floating-point range is sufficiently large to represent the values during model inference, with extreme values clamped within the range.

**Convolutions**. Convolution operations are computed in int8 to reduce computational cost. After training a NVC, all parameters are converted to integer types. Given the convolution weights $w^{\text{float}}$ and the learned step sizes $s_{iw}^{\text{float}}$ and $s_{cw}^{\text{float}}$, we convert the weights to int8 using:

$$w^{\text{int8}} = \left\lfloor \text{clip}(w^{\text{float}} \times s_{iw}^{\text{float}}, -128, 127) \right\rceil .\text{to\_int8}() \tag{11}$$

For the bias $b^{\text{float}}$, we convert it to int16 since adding a bias is a simple arithmetic as mentioned before:

$$b^{\text{int16}} = (K_b \times b^{\text{float}}).\text{to\_int16}() \tag{12}$$

And for each convolution layer, we pre-calculate two factors $s_1^{\text{int32}}$ and $s_2^{\text{int32}}$ by:

$$s_1^{\text{int32}} = (K_c \times s_{ix}^{\text{float}}).\text{to\_int32}() \tag{13}$$

$$s_2^{\text{int32}} = (K_d \times s_{cx}^{\text{float}} \times s_{cw}^{\text{float}}).\text{to\_int32}() \tag{14}$$

These factors will be used in the calculation process on the convolution layer. The hyperparameters $K_b = 2^{14}$, $K_c = 2^7$, and $K_d = 2^{24}$ are set accordingly. These preprocessing steps are performed once per NVC.

The inference process of convolution is shown in Algorithm 1. Given an input feature $x^{\text{int16}}$, we utilize the pre-calculated factor $s_1^{\text{int32}}$ to convert it to int8. Convolution is then performed in int8, with the accumulator type being int32. In integer convolution calculations, the overflow problem may cause cross-platform coding inconsistency since different platforms may behave differently for overflow. Our scheme aims to prevent overflow under **any** circumstances. Since the accumulator data

Table 5: BD-Rate (%) comparison in RGB colorspace. All frames with intra-period=–1.

| Method | UVG | MCL-JCV | HEVC B | HEVC C | HEVC D | HEVC E | Average | Coding Speed | |
| --- | --- | --- | --- | --- | --- | --- | --- | --- | --- |
| | | | | | | | | Enc. | Dec. |
| *Traditional Codecs* | | | | | | | | | |
| VTM-17.0 | 0.0 | 0.0 | 0.0 | 0.0 | 0.0 | 0.0 | 0.0 | 0.01 fps | 23.6 fps |
| HM-16.25 | 43.2 | 49.5 | 49.9 | 45.2 | 39.9 | 47.7 | 45.9 | 0.05 fps | 39.6 fps |
| *Single-Platform NVCs* | | | | | | | | | |
| DCVC-DC | 9.2 | 0.0 | 14.9 | 5.3 | −7.8 | 87.7 | 18.2 | 3.3 fps | 4.3 fps |
| DCVC-FM (fp16) | −10.4 | −1.1 | −11.2 | −26.5 | −33.7 | −12.1 | −15.8 | 5.0 fps | 5.9 fps |
| DCVC-RT (fp16) | −17.2 | −6.8 | −11.3 | −15.8 | −21.3 | −11.4 | −14.0 | 125.2 fps | 112.8 fps |
| **Our NVC (fp16)** | −30.4 | −17.0 | −24.9 | −26.2 | −30.9 | −32.7 | −27.0 | 141.1 fps | 131.0 fps |
| *Cross-Platform NVCs* | | | | | | | | | |
| DCVC-RT (int16) | −13.1 | −3.9 | −9.2 | −14.8 | −20.4 | −3.2 | −10.8 | 28.3 fps | 20.9 fps |
| **Our NVC (cross-platform)** | −14.9 | −1.2 | −8.6 | −15.0 | −20.3 | −16.9 | −12.8 | 153.0 fps | 137.3 fps |

Table 6: BD-Rate (%) comparison for our floating-point NVC in YUV420 colorspace. All frames with intra-period=–1.

| Method | UVG | MCL-JCV | HEVC B | HEVC C | HEVC D | HEVC E | Average | Coding Speed | |
| --- | --- | --- | --- | --- | --- | --- | --- | --- | --- |
| | | | | | | | | Enc. | Dec. |
| VTM-17.0 | 0.0 | 0.0 | 0.0 | 0.0 | 0.0 | 0.0 | 0.0 | 0.01 fps | 23.6 fps |
| ECM-11.0 | −20.0 | −22.1 | −22.2 | −21.2 | −20.4 | −17.2 | −20.5 | 0.002 fps | 3.4 fps |
| DCVC-FM (fp16) | −16.8 | −8.0 | −15.4 | −30.2 | −37.5 | −20.2 | −21.3 | 5.0 fps | 5.9 fps |
| DCVC-RT (fp16) | −24.0 | −14.8 | −16.6 | −21.0 | −27.3 | −22.4 | −21.0 | 125.2 fps | 112.8 fps |
| **Our NVC (fp16)** | −35.8 | −24.2 | −29.4 | −30.8 | −36.3 | −41.8 | −33.1 | 141.1 fps | 131.0 fps |

type is int32, it allows for the sum of up to 131071 products of int8 variables[3], which far exceeds the maximum possible value obtained by multiplying the number of input channels and the square of the kernel size in our model. In implementation, we split the accumulator result $a$ into its higher 16 bits $a_h$ and lower 16 bits $a_l$ to avoid overflow. Moreover, since hyperparameters $K_{a,b,c,d}$ are powers of 2, related operations are efficiently implemented via bitwise shifts. The remaining operations (addition, multiplication, and division) involve operands within the range [-32768, 32767] and will never exceed the range of int32.

**Activations**. Our integer modules use ReLU as the activation function. Since ReLU always follows a convolution in our NVC, we implement it by modifying Line 8 in Algorithm 1:

$$y^{\text{int16}} \leftarrow \text{clip}(a^{\text{int32}}, 0, 32767).\text{to\_int16}() \tag{15}$$

This implementation does not introduce any additional cost during inference.

# D  ADDITIONAL RESULTS

In this section, we provide additional experimental results.

## D.1  RGB COLORSPACE

Here we provide the rate-distortion and coding speed comparison with state-of-the-art methods. As shown in Table 10, compared to the anchor VTM, our NVC achieves an average bitrate saving of −12.8%, which outperforms cross-platform NVC DCVC-RT (int16) Jia et al. (2025). Although this is a little worse than floating-point DCVC-RT and DCVC-FM Li et al. (2024), in next section we present that the floating-point version of our NVC can surpass them by a large margin.

---

[3] $\lfloor 2147483647 / (-128 \times -128) \rfloor = 131071$

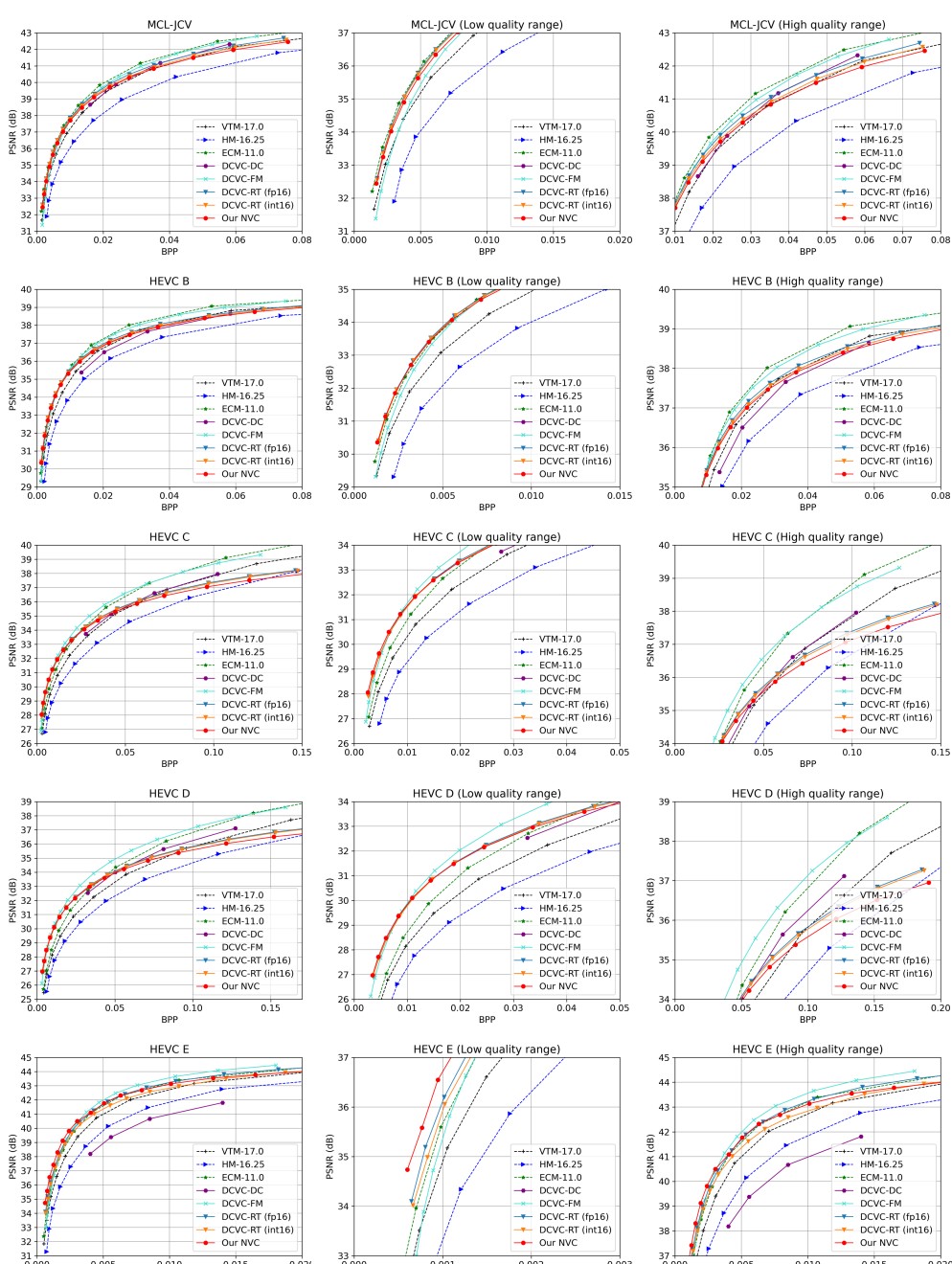

Figure 8: Rate-distortion curves on MCL-JCV and HEVC dataset. All frames are tested in YUV420 colorspace with intra-period=–1.

## D.2 FLOATING-POINT MODEL

In the paper, we primarily focus on our cross-platform NVC and present its performance. However, with the enhancements made to the memory model and module structures, we can also deliver a high-performance floating-point model outperforming state-of-the-art floating-point NVCs. To implement the floating-point model, we make two main adjustments to our cross-platform NVC: 1) we replace all integer DCBs with floating-point DCBs, and 2) we remove one DCB in the reconstruction generation module to improve decoding speed. With these modifications, we present

---

**Algorithm 1** Int8 Convolution: Inference

---

**Input:** Input feature $x^{\text{int16}}$; Convolution weight $w^{\text{int8}}$; Convolution bias $b^{\text{int16}}$; Factors $s_1^{\text{int32}}$ and $s_2^{\text{int32}}$;
    Hyperparameters $K_a$, $K_b$, $K_c$ and $K_d$.

**Output:** Output feature $y^{\text{int16}}$

    // Execute convolution

1: $x^{\text{int8}} \leftarrow x^{\text{int16}} \times s_1^{\text{int32}} \,/\, (K_a \times K_c)$

2: $a^{\text{int32}} \leftarrow \text{conv}(x^{\text{int8}}, w^{\text{int8}})$

    // Overflow prevention

3: $a_h^{\text{int32}} \leftarrow a^{\text{int32}} \gg 16$

4: $a_l^{\text{int32}} \leftarrow a^{\text{int32}} \,\&\, \text{0xFFFF}$

5: $a^{\text{int32}} \leftarrow a_h^{\text{int32}} \times s_2^{\text{int32}} \times (65536 \,/\, (K_d \,/\, K_a))$

6: $a^{\text{int32}} \leftarrow a^{\text{int32}} + a_l^{\text{int32}} \times s_2^{\text{int32}} \,/\, (K_d \,/\, K_a)$

    // Adding bias and output

7: $a^{\text{int32}} \leftarrow a^{\text{int32}} + b^{\text{int16}} \times K_a \,/\, K_b$

8: $y^{\text{int16}} \leftarrow \text{clip}(a^{\text{int32}}, -32768, 32767).\text{to\_int16}()$

---

the performance of our floating-point NVC (float) in Tables 6 and 10. In the YUV420 colorspace, our NVC achieves the best compression ratio with an average BD-Rate of $-33.1\%$, significantly outperforming $-21.3\%$ of DCVC-FM Li et al. (2024) and $-20.5\%$ of ECM ECM. Moreover, it reaches an average encoding/decoding speed of 141.1/131.0 fps, which is much faster than the previous real-time NVC DCVC-RT Jia et al. (2025). In RGB colorspace, our floating-point NVC also delivers the best compression ratio and the fastest coding speed among all floating-point NVCs, demonstrating its superior performance.

### D.3 RATE-DISTORTION CURVES

In the paper, we present the rate-distortion curves of our NVC alongside state-of-the-art codecs on the UVG dataset Mercat et al. (2020). In Fig. 8, we further provide rate-distortion curves on other datasets, including MCL-JCV Wang et al. (2016) and HEVC Flynn et al. Class B–E. At lower quality ranges, our NVC demonstrates comparable or superior performance to other methods, highlighting its exceptional efficiency. Similar to DCVC-RT Jia et al. (2025), we observe a performance decline at higher bitrates. Since our NVC employs a lightweight design for real-time coding, its model capacity is more constrained. However, as demonstrated in DCVC-RT, this performance drop can be mitigated by developing a larger model.

### D.4 RATE-DISTORTION PERFORMANCE ON DIFFERENT CONTENTS

Table 7 presents the per-sequence performance. Across diverse video content, our NVC demonstrates consistently robust performance, closely matching the behavior of our baseline DCVC-RT. It performs better on sequences with smooth motion (*e.g.*, HoneyBee) than on large motions (*e.g.*, Jockey).

Table 7: Per-sequence BD-Rate (%) in YUV420 colorspace. All frames with intra-period=–1.

| UVG (1080p) | Beauty | Bosphorus | HoneyBee | Jockey | ReadySteadyGo | ShakeNDry | YachtRide | Overall |
|---|---|---|---|---|---|---|---|---|
| DCVC-RT (fp16) | 0.0 | 0.0 | 0.0 | 0.0 | 0.0 | 0.0 | 0.0 | 0.0 |
| Our NVC (cross-platform) | 3.1 | 4.6 | –9.0 | 5.6 | 4.0 | –1.3 | 1.9 | 2.1 |

### D.5 RATE-DISTORTION PERFORMANCE ON UHD VIDEOS

Although trained only on low-resolution data, our NVC supports ultra-high-resolution (UHD) video coding. In Table 8, we evaluate its performance on UHD datasets, including JVET Class A1 and A2. Compared to the chip-accelerated traditional codec NVEnc-HEVC, our NVC achieves 70.4% and

63.2% bit savings on Class A1 and A2, respectively. This is comparable to DCVC-RT, demonstrating its effectiveness on UHD content.

Notably, we do not compare against more advanced traditional codec standards like H.266/VTM and ECM due to their significant computational demands. To our knowledge, official test results for low-delay settings on Class A1 and A2 are not yet available, and running them locally would require months of processing time. To better illustrate our UHD performance, we additionally provide comparison results on the 1080p Class B dataset. Our NVC shows similar bit savings on both 1080p and UHD content, highlighting its robustness across resolutions.

Table 8: BD-Rate (%) in YUV420 colorspace on UHD datasets. All frames with intra-period=–1.

| Dataset | Class A1 | Class A2 | Class B |
|---|---|---|---|
| NVEnc-HEVC | 0.0 | 0.0 | 0.0 |
| DCVC-RT (fp16) | –72.8 | –64.5 | –69.0 |
| Our NVC (cross-platform) | –70.4 | –63.2 | –68.0 |

Table 9: Encoding / decoding speed (frame per seconds, fps) on various resolutions and devices.

| Resolution | 3840×2160 | 1920×1080 | 1280×720 |
|---|---|---|---|
| A100 | 46.5 / 41.7 | 153.0 / 137.3 | 257.0 / 205.2 |
| 2080Ti | 16.0 / 15.4 | 56.9 / 54.3 | 112.4 / 102.8 |

### D.6 CODING SPEED ANALYSIS

In Table 9, we further present the coding speed of our cross-platform NVC on the NVIDIA A100 and RTX 2080Ti GPUs. Our NVC achieves an impressive 40 fps for 4K video coding on the A100, and more than 50 fps for 1080p content on the consumer-grade 2080Ti. These results demonstrate the efficiency of our NVC.

### D.7 VISUAL COMPARISON

For a more comprehensive understanding of the reconstruction quality, we provide the decoded videos of VTM VTM, DCVC-RT Jia et al. (2025), and our NVC in the supplementary files. Specifically, we use FFmpeg ffm to compress the decoded frames at a very high quality (crf=10) to avoid introducing additional compression artifacts. These videos are stored at 30 fps. We provide two sequences: 1) videoSRC28_1920x1080_30 from MCL-JCV, and 2) Cactus_1920x1080_50 from HEVC Class B, with the first 300 frames retained to stay within size limits. Compared to VTM, our NVC reconstructs details more effectively, especially for regions like fonts. In comparison to DCVC-RT, our NVC better preserves fine details and handles scene changes more robustly.

Table 10: Visual comparison details. The bpp (bits per pixel) / PSNR are presented for each sequence.

| Methods | VTM-17.0 | DCVC-RT | Our NVC |
|---|---|---|---|
| videoSRC28 | 0.0050 / 41.47 | 0.0050 / 42.36 | 0.0049 / 42.35 |
| Cactus | 0.0076 / 33.80 | 0.0075 / 34.57 | 0.0073 / 34.61 |

## E CLAIM OF USING LLMS IN PAPER WRITING

Large Language Models (LLMs) were utilized exclusively to refine the clarity and readability of the text. No assistance from LLMs was involved in literature retrieval, idea generation, or any other stage of the writing process.

