# OpenReview forum: "Integer-Centric Neural Video Compression"
_ICLR.cc/2026/Conference — Submitted to ICLR 2026_

### Official Review · Reviewer_RwNV · 2025-10-29

**Soundness:** 3
**Presentation:** 3
**Contribution:** 4
**Rating:** 6
**Confidence:** 5

**Summary:**

The paper proposes a high-performance cross-platform neural video codec (NVC) that operates fully in the integer domain, ensuring bit-exact consistency across different hardware platforms. Unlike prior “floating-point-centric” methods that quantize pretrained float models, this work introduces an integer-centric training pipeline, training the integer model from scratch. Furthermore, they introduce a memorized temporal modeling mechanism. The proposed model achieves an average 20% bitrate reduction compared to H.266/VTM while maintaining an encoding/decoding speed of 153.0/137.3 fps for 1080p video.

**Strengths:**

The cross-platform coding consistency issue is of great significance for end-to-end video compression. In this paper, the authors propose the idea of training an INT16 model from scratch. Unlike other model quantization tasks, LSQ cannot be directly and effectively applied to compression tasks. To address this, the authors introduce a novel method called Multiply-twice integerization to overcome the problem, which is simple yet highly efficient. With the help of multiply-twice integerization and another proposed memorized temporal modeling, their method exhibits good RD performance at a very fast speed.

**Weaknesses:**

This paper is well-written overall. Nevertheless, it still has some weaknesses.
1. The issue of training instability lacks sufficient analysis. It should provide statistical data on key variables in the training process and compare them to demonstrate that the training effectiveness has indeed improved with Multiply-twice Integerization.
2. The correlation between Multiply-twice Integerization and the FP16 (or FP32) model requires further analysis. For example, is it only effective for INT8 models but not for FP16 (or FP32) models? Could this design also provide performance benefits when applied to an FP16 (or FP32) model trained from scratch?
3. Figure 5 should also account for the variation in decoding speed or complexity.
4. Lack of experiments with Intra Period (IP) 32, following the setting of other mainstream methods like DCVC-FM.
5. Lack of detailed training costs, like the number and version of the GPUs, and the days of training time.
6. Lack of an exact formulation of the loss function.
7. Typos: Line 161, "Here, v is is"; Line 244, "he gradient".

**Questions:**

1. Is there any multi-stage training strategy similar to DCVC-TCM? Will the training code be released?
2. It seems that INT8 acceleration boosts speed at the cost of RD performance, whereas in-loop integerization and memorized temporal modelling sacrifice speed for better compression efficiency. Could you provide a detailed breakdown of the specific gains and penalties introduced by each component?
3. In Line 971, you remove one DCB in the FP16 model. This seems to be the only change relative to DCVC-RT (FP16) that can increase speed. How much speed-up does it actually yield, and what performance loss does it incur? Is that the main factor that makes the FP16 model faster than DCVC-RT in Table 5?
4. During the training of the entropy model, is noise injection adopted to simulate quantization error? In integer model training, what distinct statistical characteristics would the quantization errors exhibit?
5. Line 251: "If it is applied in LSQ, the coupled gradient may cause an incorrect updating of s". Please provide a more detailed explanation of "incorrect updating".
6. Table 1 shows In-Loop Decoding obtains 3.4% gains, while Table 4 shows In-Loop integerization obtains (30.6-15.4)% gains in C3. What's the difference between In-Loop Decoding and In-Loop integerization? Are they the same concept?
7. In Line 836, what is the "extensive kernel fusions"?

---

> ### Author Response · Authors · 2025-11-17
> **Rebuttal to Reviewer RwNV, Part1**
>
> Thanks for your careful review and the positive feedback on our paper. We provide clarifications on your questions below.
>
> **W1 \& W2: Analysis and future illustration on multiply-twice integerization.**
>
> Thanks for your suggestion. To statistically analysis the stability, we train integer NVCs from scratch using learned step size (LSQ) and multiply-twice quantization, each with three independent trials. We report both the BD-Rate and the maximum scaling factor (i.e., $\max(1/|s|, |s|)$ for LSQ and $\max(|s_i|, |s_c|)$ for ours) across training epochs. As shown in the below table, LSQ consistently produces large scaling values (up to $10^4$–$10^6$), often leading to divergence. In contrast, our method keeps scaling factors within 2–4, enabling stable training and good BD-Rate performance.
>
> |BD-Rate (%)|epoch 10|epoch 20|epoch 30|epoch 40|epoch 50|epoch 60|epoch 70|epoch 80|epoch 90|
> |-|-|-|-|-|-|-|-|-|-|
> |LSQ, trial 1|$>10^5$|723|$>10^5$|4330|NAN|NAN|NAN|NAN|NAN|
> |LSQ, trial 2|$>10^5$|579|$>10^5$|$>10^5$|$>10^5$|9427|549|$>10^5$|$>10^5$|
> |LSQ, trial 3|$>10^5$|2480|23314|4171|650|543|2413|$>10^5$|$>10^5$|
> |Multiply-twice, trial 1|4068|135|57|56|50|36|10|0.8|0.0|
> |Multiply-twice, trial 2|3866|130|57|51|52|44|14|3.0|1.2|
> |Multiply-twice, trial 3|4106|133|58|54|43|37|21|2.3|1.1|
>
> |Max scaling value|epoch 10|epoch 20|epoch 30|epoch 40|epoch 50|epoch 60|epoch 70|epoch 80|epoch 90|
> |-|-|-|-|-|-|-|-|-|-|
> |LSQ, trial 1|$10^1$|$10^3$|$10^5$|$10^4$|NAN|NAN|NAN|NAN|NAN|
> |LSQ, trial 2|$10^1$|$10^2$|$10^6$|$10^4$|$10^4$|$10^3$|$10^5$|$10^6$|$10^6$|
> |LSQ, trial 3|$10^1$|$10^4$|$10^5$|$10^4$|$10^4$|$10^3$|$10^6$|$10^4$|$10^4$|
> |Multiply-twice, trial 1|2.1|3.0|3.3|3.8|4.1|4.1|4.1|4.0|4.0|
> |Multiply-twice, trial 2|2.1|3.0|3.3|3.9|4.0|4.0|4.0|4.0|4.0|
> |Multiply-twice, trial 3|2.0|3.0|3.3|3.9|4.0|4.1|4.1|4.1|4.1|
>
> Regarding the relationship between multiply-twice integerization and floating-point models, multiply-twice technique is primarily effective for integer models and not for floating-point models. Similar to LSQ, the multiply-twice method is specifically designed to handle value scaling during integerization, which is unnecessary for floating-point models.
>
> **W3: Complexity analysis in Figure 5.**
>
> We provide complexity statistics for different memory-channel sizes and will incorporate this information into the revised Figure 5.
>
> |ChannelSize|0|128|256|384|512|
> |-|-|-|-|-|-|
> |kMACs/pixel|143.3|150.8|170.1|204.3|250.2|
> |Parameters|15.7M|16.6M|19.0M|23.2M|28.9M|
>
> **W4: Results on intra-period of 32.**
>
> Thanks for the suggestion. According to DCVC-FM, an intra-period of -1 achieves a higher compression ratio than 32 and is the standard defined by the traditional committee [11], which is why we primarily test -1 in the paper. For completeness, we provide results with an intra-period of 32 below and will include them in the revision.
>
> |Method|UVG|MCL-JCV|HEVC B|HEVC C|HEVC D|HEVC E|Average|
> |-|-|-|-|-|-|-|-|
> |VTM|$0.0$|$0.0$|$0.0$|$0.0$|$0.0$|$0.0$|$0.0$|
> |**Our NVC (fp16)**|$-23.5$|$-15.9$|$-15.7$|$-16.4$|$-28.3$|$-22.0$|$-20.3$|
> |**Our NVC (cross-platform)**|$-11.1$|$-3.0$|$-2.1$|$-7.1$|$-20.7$|$-13.7$|$-9.6$|
>
> [1] Frank Bossen et al. Common test conditions and software reference configurations. In JCTVC-L1100, 2013
>
> **W5: Training costs.**
>
> Our model was trained on 2 H100 GPUs for approximately 8 days.
>
> **W6: Formulation of loss function.**
>
> Our loss function is the same as DCVC-FM: $L=R+\lambda(k\cdot D_{YUV}+(1-k)\cdot D_{RGB})$.
>
> **W7: Typos.**
>
> Thanks, we will revise them.

---

> > ### Author Response · Authors · 2025-11-17
> > **Rebuttal to Reviewer RwNV, Part2**
> >
> > **Q1: Training strategy and code**
> >
> > Since our NVC does not have a motion-estimation-motion-compensation module, its training strategy is much simpler than DCVC-TCM. We warm-up model with only distortion loss, then train it with rate-distortion loss. We will release the training scripts upon acceptance.
> >
> > **Q2: Breakdown of specific gains of each component.**
> >
> > We categorize the changes of components into two parts for a breakdown analysis:
> >
> > - Network structure improvements. It includes the addition of the memory module and redistribution of parameters across modules. These changes enhance both speed and compression efficiency, as discussed in our response to Q3.
> >
> > - Model integerization. Increasing the number of integerized modules generally boosts speed but can reduce compression efficiency. Consequently, the floating-point model is the slowest but achieves the best RD performance; the fully integerized model is the fastest with lower RD; and the partially integerized in-loop version achieves intermediate speed and RD performance.
> >
> > **Q3: Analysis on the speed improvement of FP16 models.**
> >
> > The removal of one DCB in Line 971 is relative to our integer model, not to DCVC-RT. Compared to DCVC-RT, our NVC also incorporates the memory module and redistributes parameters across modules to optimize overall performance (see Figure 6). Therefore, the observed speed-up and performance improvements result from this holistic network adjustment rather than the single DCB removal.
> >
> > **Q4: Noise injection in entropy model and statistical characteristics of quantization errors.**
> >
> > We add uniform noise when estimating entropy and use straight-through estimation (STE) to quantize the encoded features when feeding them to the decoder.
> >
> > For one-stage integer training, since it does not rely on a reference floating-point model, the notion of “quantization error” of a ground-truth floating-point value does not apply. If you refer to the "rounding error", it is naturally bounded within $[-0.5,0.5)$, with its exact distribution determined by the individual feature values.
> >
> > **Q5: Further explanation on gradient when adopt Clipping Gradient Pass on LSQ.**
> >
> > In multiply-twice integerization, $\hat{v} = \left\lfloor \text{clip}(v\times s_{i}, -Q_N, Q_P)\right\rceil\times s_{c}$, we allow gradients to flow through the decoupled $s_i$ when the gradient-pass condition is satisfied. The gradient is $\frac{\partial \hat{v}}{\partial s_i}=v\times s_{c}$.
> >
> > In LSQ, $\hat{v} = \left\lfloor \text{clip}(v/s, -Q_N, Q_P)\right\rceil\times s$, there is a single step size $s$. If one tries to mimic the same gradient $\frac{\partial \hat{v}}{\partial (1/s)}=v\times s$ by treating $1/s$ and $s$ separately, it introduces bias because the gradient w.r.t. the multiplied $s$ is ignored.
> >
> > For example, suppose $v/s > Q_P$ and the gradient aims to reduce $v/s$ (gradient pass condition satisfied). Updating $s$ based on this biased gradient may actually increase $s$. However, a larger $s$ leads to a larger $\hat{v}$ resulting in a greater difference from $v$, i.e., a larger quantization error is caused by the biased gradient.
> >
> > **Q6: In-Loop Decoding and In-Loop integerization.**
> >
> > These terms refer to different scopes.
> >
> > - In-Loop Decoding (Table 1): Refers only to the decoder. The decoding module includes in-loop decoding modules (decoder, memory, feature extractor, etc) and out-loop reconstruction generation module. We compared integerizing the out-of-loop reconstruction (3.4\% loss) versus keeping it in float.
> >
> > - In-Loop integerization (Table 4): Refers to the full codec (encoder + decoder). We compared integerizing the out-of-loop modules including encoder on both sides (C2, 30.6\%) versus keeping them in float (C3, 15.4\%).
> >
> > **Q7: Explanation on kernel fusions.**
> >
> > Kernel fusion is a technique that combines multiple GPU operations (e.g., convolution, activation, and element-wise operations) into a single fused kernel. This reduces memory I/O overhead and improves inference speed, and is commonly used in memory-bound operations like attention (Flash Attention).

---

> > > ### Comment · Reviewer_RwNV · 2025-11-25
> > >
> > > Thanks for your responses. They have addressed most of my concerns. However, there are still some issues that require further explanation from the authors.
> > >
> > > **W3:**
> > >
> > > The table in W3 needs to include encoding and decoding speeds, as kMACs sometimes do not accurately reflect the actual coding speed.
> > >
> > > **Q2&Q3:**
> > >
> > > What specifically does "Network structure improvements" in Q2 refer to? Given its effect, this is a noteworthy change. What are the improvements in RD performance and changes in complexity/speed it brings? Please provide specific experimental results.
> > >
> > > **Q5:**
> > >
> > > It still lacks a more intuitive explanation. As the author mentioned, Clipping Gradient Pass and multiply-twice integerization are closely related. Therefore, this section requires more elaboration. For example, similar to the derivation from equation (5) to (8), the author could extend Clipping Gradient Pass (or similar ideas) to LSQ—i.e., derive equation (2) into a new form and compare it with equation (8), before explaining what the bias is.

---

> ### Author Response · Authors · 2025-11-26
> **Further Reply to Reviewer RwNV**
>
> Thanks for your time in following up with the discussion. We are glad that most of your concerns were addressed in the rebuttal. For the remaining points, we provide additional clarification below.
>
> **W3**
>
> We provide the coding speed results below. As the channel size decreases, the coding speed increases accordingly.
>
> | Channel Size | 0 | 128 | 256 | 384 | 512 |
> |--------------|---|-----|-----|-----|-----|
> | kMACs/pixel  | 143.3 | 150.8 | 170.1 | 204.3 | 250.2 |
> | Parameters   | 15.7M | 16.6M | 19.0M | 23.2M | 28.9M |
> | Enc./Dec. Speed (fps, 1080p, A100) | 194.0 / 172.8 | 187.4 / 165.3 | 174.0 / 153.7 | 153.0 / 137.3 | 135.4 / 123.8 |
>
> **Q2**
>
> “Network structure improvements” refer to the introduction of memory and the resulting structural adjustments, including:
> - Adding a memory module with three DCBs;
> - Reducing the feature extractor to four DCBs, since memory enhances context learning;
> - Reducing the reconstruction generation module’s channel size from 320 to 256, as memory also improves decoding feature learning.
>
> Since all these modifications stem from the memory mechanism, we jointly ablate them as “memorized temporal modelling” in Table 4. These changes collectively yield about a 14% compression improvement.
>
> For the floating-point models, we further remove one DCB from the reconstruction generation module. With all these enhancements, our floating-point NVC reaches a BD-Rate of −27.0% and encoding/decoding speeds of 141.1 / 131.0 fps, surpassing DCVC-RT (fp16) with a BD-Rate of −14.0% and speeds of 125.2 / 112.8 fps.
>
> The text version is less detailed regarding the architectural modifications. If you are interested in the full design, please refer to the framework illustration in Figure 6.
>
> **Q5**
>
> In LSQ,  $\hat{v}=\left\lfloor \mathrm{clip}(\frac{v}{s},-Q_N,Q_P)\right\rceil \times s$, we explained the clipping gradient pass case for $\frac{v}{s} > Q_P$. The case for $\frac{v}{s} < -Q_N$ is analogous.
>
> LSQ naturally defines the gradient when $\frac{v}{s} > Q_P$ as $\frac{\partial \hat{v}}{\partial s} = Q_P$ (Equation 2). This increases $s$, reducing $\frac{v}{s}$ so that it falls back below $Q_P$, which matches LSQ’s intended behavior.
>
> However, if clipping gradient pass is enforced, then
>
> $
> \frac{\partial \hat{v}}{\partial s} =
> \begin{cases}
> -\frac{v}{s} + \left\lfloor \frac{v}{s} \right\rfloor, & \text{if} -\frac{v}{s} + \left\lfloor \frac{v}{s} \right\rfloor >0, \\\\
> 0, & \text{if} -\frac{v}{s} + \left\lfloor \frac{v}{s} \right\rfloor \le 0.
> \end{cases}
> $
>
> This deviates from the original derivation and, importantly, prevents updating $s$ when the expression is negative. It introduces the “bias” previously mentioned.
>
> In early stage of this work, we performed experiments to train integer models from scratch under LSQ with clipping gradient pass. Without the clipping gradient pass, LSQ diverged with >500% loss inflation (Table 5). With clipping pass, performance degraded even further and often failed to converge.
>
> We hope these explanations address your remaining concerns. If there are any additional questions, we would be happy to continue the discussion.

---

### Official Review · Reviewer_dbJf · 2025-10-30

**Soundness:** 3
**Presentation:** 3
**Contribution:** 3
**Rating:** 6
**Confidence:** 5

**Summary:**

This paper presents the first fully 8-bit integer neural video codec (NVC) trained end-to-end without any floating-point initialization. The proposed framework achieves real-time 1080p compression and ~20% bitrate reduction compared to H.266/VTM, while ensuring bit-exact cross-platform consistency. A novel multiply-twice quantization strategy and a memory-augmented temporal module jointly mitigate the stability–accuracy trade-off inherent to low-precision video compression.

**Strengths:**

1. The method completely abandons the conventional “float → quantize” pipeline and successfully trains an integer-only codec from scratch, representing a pioneering step for low-precision end-to-end video compression.
2. The proposed integerization removes division operations, suppresses gradient explosion, and maintains stable optimization for long GOP sequences in 8-bit precision.
3. The lightweight memory module effectively compensates for capacity loss due to quantization, providing up to 17% additional bitrate reduction for the integer model.
4. The work demonstrates thorough engineering validation, including bit-exact verification across multiple GPUs and Intel iGPUs, and extensive runtime benchmarks from 720p to 4K.
5. The codec achieves superior rate-distortion (RD) efficiency and runtime compared to both VTM and existing cross-platform NVCs, highlighting its strong practical significance.

**Weaknesses:**

1. Limited theoretical justification. No formal convergence analysis or error bound is provided for the multiply-twice scheme; the “division-free ⇒ stability” claim remains heuristic.
2. RD curves flatten at higher bitrates, suggesting limited representational capacity under strict 8-bit constraints.
3. Ablation studies are insufficient. The impact of network width, memory channel count, and QP granularity is not individually evaluated; no direct comparison between 8-bit and 16-bit integer models under identical settings.

**Questions:**

1. The current consistency verification is limited to desktop GPUs and a single Intel iGPU. Additional validation on ARM-based devices, mobile DSPs, or heterogeneous compiler stacks would be valuable to confirm true cross-platform reproducibility.
2. The quantization-error bounds and convergence guarantees of the multiply-twice integerization remain unclear. A formal derivation or at least an empirical characterization of the error distribution would strengthen the technical soundness of the method.
3. The paper could further explore more advanced temporal memory mechanisms, such as causal self-attention or LSTM-based recurrent modules, to assess whether they offer improved rate–distortion trade-offs or different complexity behaviors.
4. Verification of bit-exactness on additional hardware backends—such as ARM NEON, Apple A-series, or Qualcomm Hexagon, where 8-bit multiply–accumulate overflow handling differs—would provide stronger evidence of platform independence.
5. The observed high-rate degradation suggests limited representational capacity under strict 8-bit constraints. Discussion of potential remedies, including model scaling, mixed-precision schemes, or adaptive bit-width allocation, would enhance completeness.
6. A controlled comparison against a re-implementation of DCVC-RT using the same 8-bit quantization-aware training setup would clarify the relative performance gap and further substantiate the claimed improvements.

---

> ### Author Response · Authors · 2025-11-17
> **Rebuttal to Reviewer dbJf, Part1**
>
> Thanks for your careful review and the positive feedback on our paper. We provide clarifications on your questions below.
>
> **W1 \& Q2: Providing convergence analysis as theoretical justification.**
>
> Formal convergence guarantees or strict error-bar derivations for learning-based quantization remain challenging. For instance, our baseline Learned Step Quantization (LSQ) also does not provide such theoretical proofs. To better clarify the theoretical motivation of our proposal, we provide analysis from two complementary perspectives:
>
> (1) LSQ can be viewed as a special instance of our multiply-twice formulation, and therefore the theoretical error bound of multiply-twice quantization is at least no worse than LSQ. Given LSQ: $\hat{v} = \left\lfloor \text{clip}(v/s, -Q_N, Q_P)\right\rceil\times s$ and multiply-twice: $\hat{v} = \left\lfloor \text{clip}(v\times s_{i}, -Q_N, Q_P)\right\rceil\times s_{c}$, we find LSQ is a special case where $s_{i}=1/s$ and $s_c=s$ are coupled form. Thus, multiply-twice generalizes LSQ, and its theoretical quantization error bound cannot be higher than that of LSQ.
>
> (2) Empirical analysis. To further characterize stability, we train integer NVCs from scratch using LSQ and multiply-twice quantization, each with three independent trials. We report both the BD-Rate and the maximum scaling factor (i.e., $\max(1/|s|, |s|)$ for LSQ and $\max(|s_i|, |s_c|)$ for ours) across training epochs. As shown in the below table, LSQ consistently produces large scaling values (up to $10^4$–$10^6$), often leading to divergence. In contrast, our method keeps scaling factors within 2–4, enabling stable training and good BD-Rate performance.
>
> |BD-Rate (%)|epoch 10|epoch 20|epoch 30|epoch 40|epoch 50|epoch 60|epoch 70|epoch 80|epoch 90|
> |-|-|-|-|-|-|-|-|-|-|
> |LSQ, trial 1|$>10^5$|723|$>10^5$|4330|NAN|NAN|NAN|NAN|NAN|
> |LSQ, trial 2|$>10^5$|579|$>10^5$|$>10^5$|$>10^5$|9427|549|$>10^5$|$>10^5$|
> |LSQ, trial 3|$>10^5$|2480|23314|4171|650|543|2413|$>10^5$|$>10^5$|
> |Multiply-twice, trial 1|4068|135|57|56|50|36|10|0.8|0.0|
> |Multiply-twice, trial 2|3866|130|57|51|52|44|14|3.0|1.2|
> |Multiply-twice, trial 3|4106|133|58|54|43|37|21|2.3|1.1|
>
> |Max scaling value|epoch 10|epoch 20|epoch 30|epoch 40|epoch 50|epoch 60|epoch 70|epoch 80|epoch 90|
> |-|-|-|-|-|-|-|-|-|-|
> |LSQ, trial 1|$10^1$|$10^3$|$10^5$|$10^4$|NAN|NAN|NAN|NAN|NAN|
> |LSQ, trial 2|$10^1$|$10^2$|$10^6$|$10^4$|$10^4$|$10^3$|$10^5$|$10^6$|$10^6$|
> |LSQ, trial 3|$10^1$|$10^4$|$10^5$|$10^4$|$10^4$|$10^3$|$10^6$|$10^4$|$10^4$|
> |Multiply-twice, trial 1|2.1|3.0|3.3|3.8|4.1|4.1|4.1|4.0|4.0|
> |Multiply-twice, trial 2|2.1|3.0|3.3|3.9|4.0|4.0|4.0|4.0|4.0|
> |Multiply-twice, trial 3|2.0|3.0|3.3|3.9|4.0|4.1|4.1|4.1|4.1|
>
> **W2 \& Q5: High-bitrates performance drop due to low precision.**
>
> We agree that strict 8-bit integerization imposes a representational ceiling. This is a common phenomenon in all integer NVCs: for example, DCVC-RT shows a 0.1 dB loss even at high Int16 precision, while MobileNVC experiences over 1 dB loss under Int8. In comparison, our NVC demonstrates that Int8 quantization incurs only around 0.1 dB PSNR drop at high rates, mitigating this limitation more effectively than prior work.
>
> Several remedies may further improve performance at high bitrates.
> - Scaling model parameters can theoretically recover floating-point performance with only polylogarithmic cost [1].
> - Instead of uniform Int8 precision across all modules, adaptive or mixed-precision schemes could exploit module-specific quantization sensitivity.
> - Since we prove that an integer NVC can be trained and designed from scratch, integer-centric architectures and algorithms offer additional opportunities to enhance high-rate performance. These points will be incorporated in the revision.
>
> [1] On the Universal Approximability and Complexity Bounds of Quantized ReLU Neural Networks
>
> **W3: About individual ablation studies.**
>
> We analyze the ablation studies as follows:
>
> - Memory-channel size and module width: Individual studies are provided in Figure 5, where compression ratio consistently improves as the channel count increases. For completeness, the corresponding complexity metrics are reported below:
>
> |ChannelSize|0|128|256|384|512|
> |-|-|-|-|-|-|
> |kMACs/pixel|143.3|150.8|170.1|204.3|250.2|
> |Parameters|15.7M|16.6M|19.0M|23.2M|28.9M|
>
>
> - QP granularity: Section 4.4 (line 338) evaluates QP-aware step sizes. Removing finer-grained QP-dependent step sizes results in 15\% performance loss, highlighting the importance of fine QP granularity.
>
> - Bit width for quantization: We evaluate models with identical settings across bit widths from Int4 to Int16. As expected, higher precision consistently improves RD performance. For example, Int16 outperforms Int8 by 9.7\%, while Int4 exhibits substantial degradation of more than 400\%.
>
> |BitWidth|Int4|Int6|Int8|Int12|Int16|
> |-|-|-|-|-|-|
> |BD-Rate|437.4%|4.4%|0%|-6.5%|-9.7%|

---

> > ### Author Response · Authors · 2025-11-17
> > **Rebuttal to Reviewer dbJf, Part2**
> >
> > **Q1 \& Q4: Cross-platform verification.**
> > In our NVC, cross-platform consistency is strictly ensured. We will analyze it in following aspects.
> >
> > - **Theoretical level**: Full integer decoding **strictly ensures** calculation consistency. This principle has been widely adopted and verified in traditional video codecs such as H.264/AVC, H.265/HEVC, and H.266/VVC. As long as deterministic integer operations are performed, the outputs are provably identical across hardware.
> >
> > - **Engineering level**: Our NVC is designed to avoids overflow to strictly ensure bitwise consistency. We agree with the reviewer that different platforms may handle overflow differently, and codec engineers typically address this by avoiding overflow.
> > For example, during HEVC standardization process, JCTVC-F537 noted that “In extreme cases values up to +33150 may be computed and they exceed the signed 16-bit range (max +32767).” The accepted solution is that “Fix the software by adding a negative offset to the values during intermediate steps to make sure they always hold in the signed 16-bit range (this has no normative impact).” Similarly, in Section 4.4 (line 345), we also investigated the potential overflow in NVC. After careful design, we have made sure that there is **no overflow risk** in our integer neural codec.
> >
> > - **Empirical level**: We have verified cross-platform consistency on multiple **NVIDIA GPUs and Intel Graphics** in the paper. To better support it, we further verify the consistency across **CPU and GPU devices**. We hope the reviewers understand that implementing an NVC across different hardware backends is primarily an engineering task with substantial effort. For example, deploying on a Qualcomm Hexagon DSP requires converting the PyTorch model to ONNX and then mapping it to the device using Qualcomm SNPE or the Hexagon SDK. We believe such implementations are more appropriate for codec engineers rather than researchers.
> >
> > If you think there are any points that potentially cause inconsistency, we are happy to further discuss or verify it.
> >
> > **Q3: Discussion on more advanced memory mechanism.**
> >
> > Thanks for the valuable suggestions. More advanced temporal mechanisms, such as causal self-attention or deeper recurrent networks, are indeed promising directions. Currently, we adopt a simple RNN-like memory primarily for its efficiency. It only requires a lightweight concatenation of memory and features, keeping the computational and memory overhead low. In contrast, mechanisms like self-attention would significantly increase complexity. Future work will explore these architectures while maintaining the integer inference requirement.
> >
> > **Q6: Implement Int8 on DCVC-RT.**
> >
> > In our ablation study (Table 4), DCVC-RT serves as the anchor. Across settings from A to C3, the model corresponds to DCVC-RT, demonstrating that applying the same 8-bit quantization-aware training results in only ~15\% performance loss. This is substantially lower than the ~80\% loss observed in MobileNVC.

---

### Official Review · Reviewer_94Vq · 2025-10-31

**Soundness:** 3
**Presentation:** 2
**Contribution:** 3
**Rating:** 4
**Confidence:** 4

**Summary:**

The paper proposes an integer-centric approach to building a cross-platform neural video codec, extending DCVC-RT. It introduces a multiply-twice strategy that avoids divisions by using decoupled step sizes inspired by Learned Step Size Quantization, and adds a memorized temporal modeling mechanism to recover capacity under strict integer arithmetic. As reported, the method achieves competitive coding efficiency while ensuring bit-exact cross-platform decoding, thereby addressing the cross-platform incompatibility issue.

**Strengths:**

The integer-centric training pipeline is a strong contribution. It departs from conventional workflows that rely on quantization-aware training (QAT) or post-training quantization (PTQ), treating integerization as a first-class objective during model design and optimization.

Cross-platform determinism for learned codecs is a key deployment barrier in real-world settings, and this work addresses it directly. Achieving real-time throughput further strengthens the practical relevance of the approach. Additionally, focusing on integerization not only mitigates cross-platform round-off errors but also points toward more energy-efficient dedicated designs, given the lower power consumption of integer versus floating-point operators. This is especially important for battery-powered devices, as edge devices account for a significant share of video consumption.

The memorized temporal modeling block is a well-motivated addition. Integrated into the DCVC-RT backbone, it provides a principled way to boost temporal prediction beyond motion compensation and, per the reported ablations, contributes materially to the observed rate–distortion gains over a no-memory baseline. Further, the multiply-twice strategy is a meaningful contribution that extends LSQ-style quantization. By separating these scales and enforcing a fixed, division-free compute path, the method improves training stability, preserves dynamic range where needed, and provides clearer fixed-point semantics for deterministic deployment. The reported ablations indicate that this design materially contributes to the observed rate–distortion gains.

The proposed method achieves competitive, and in several cases superior, rate–distortion performance compared to state-of-the-art DCVC-based frameworks under the reported evaluation protocol, indicating that the design choices translate into efficient coding gains.

**Weaknesses:**

**Related work coverage.** The paper omits a key ICLR reference on *integer networks for compression* [R1], which directly targets bit-exact decoding. Beyond being a relevant citation, this work is foundational, since it explicitly surfaced the cross-platform decoding consistency problem for learned compression and demonstrated integerization as a viable solution to achieve bit-exact behavior across heterogeneous hardware. Please cite this paper and position the present contributions relative to it. Additionally, the experimental comparison is largely confined to DCVC's extensions. Please broaden the baselines to include representative non-DCVC NVCs and report head RD results under the same protocol. Note that there are NVCs in the literature that report rate–distortion performance meeting or surpassing DCVC-FM. Restricting comparisons to DCVC variants makes it difficult to assess competitiveness against the broader SOTA methods. If the scope is intentionally restricted, clarify the selection criteria and justify why comparisons outside the DCVC family are omitted.

**Practical deployment.** As acknowledged in the conclusion, the proposed NVC is not yet suitable for edge devices with tight computational and energy constraints. A key factor appears to be the 384-channel design of the memorized temporal modeling block, which likely increases compute, activation size, and memory traffic. To clarify deployment implications, please report the computational cost as MACs per pixel (MACs/px) and, for low-precision arithmetic, bit-operations per pixel (BOPs/px), and break these out specifically for the memorized temporal module.

**Quantization precision.** The paper largely fixes the integer path at 8-bit and does not examine how coding efficiency and complexity trade off under more aggressive (e.g., 6- or 4-bit) or less aggressive (e.g., 10-, 12-, 16-bit) precisions. Presenting experiments that vary bit widths and report RD impact alongside complexity metrics would enhance the paper. In addition, with a less aggressive precision, the memorized temporal modeling block might admit fewer channels while preserving most of the gains, yielding a better Pareto between complexity and coding efficiency.

**CPU support rationale is too vague.** The claim that CPU support is excluded due to “significant engineering” needs specificity. This justification is difficult to follow, since integer operations are ubiquitous on CPUs. All modern processors using the x86 architecture natively support integer arithmetic, and even Intel’s 8086, released in 1978, implemented integer arithmetic. For reproducibility and scope clarity, please disclose what is missing and what is the specific challenge for implementing integer operations on CPU. As written, it is ambiguous whether the barrier is that you have not implemented a CPU kernel yet, or that you did but could not achieve bit-exact cross-platform consistency.

**UHD comparisons practicality.** The claim that Class A1/A2 (UHD) would take “months” seems overstated, mainly for VVC/VTM. Additionally, running VVenC [R4] at an appropriate preset, or temporally sampling shorter sequence segments for ECM, would improve the experiments quality in this scenario.

### Minor Issues
**Typos, article/subject-verb slips, and style inconsistencies.** The paper is clear overall but would benefit from a careful proofread to fix numerous small slips and improve polish: clean up article/subject–verb agreement (“a/an”, plural vs singular), remove duplicated words (e.g., “is is the quantized”), and choose one spelling (“modeling” vs “modelling”).

**Inadequate references.** The citations for H.264/AVC, H.265/HEVC, and H.266/VVC point to HM/VTM source repos [R2], [R3] (or for the Overview paper in the case of H.264/AVC) rather than the normative specs. For claims relative to “H.26x,” please cite the ITU-T/ISO standards [R5], [R6], [R7].

**Figure 2 not referenced/discussed.** Figure 2 appears in the paper but is never cited or explained in the main text; please add an explicit reference (e.g., “see Fig. 2”) at first mention and briefly state its takeaway and relevance.

## References
- **[R1]** Ballé, J., Johnston, N., & Minnen, D. (2019). *Integer Networks for Data Compression with Latent-Variable Models.* International Conference on Learning Representations (ICLR).
- **[R2]** HM: <https://vcgit.hhi.fraunhofer.de/jvet/HM>
- **[R3]** VTM: <https://vcgit.hhi.fraunhofer.de/jvet/VVCSoftware_VTM>
- **[R4]** VVEnc: <https://github.com/fraunhoferhhi/vvenc>
- **[R5]** ITU-T Recommendation H.264 and ISO/IEC 14496-10 (MPEG-4 AVC): *Advanced video coding for generic audiovisual services*.
- **[R6]** ITU-T Recommendation H.265 and ISO/IEC 23008-2 (MPEG-H Part 2): *High efficiency video coding*.
- **[R7]** ITU-T Recommendation H.266 and ISO/IEC 23090-3 (MPEG-I Part 3): *Versatile video coding*.

**Questions:**

**Question 1:** In your ablation, coding efficiency improves as the number of channels in the memorized temporal module increases. For the 0-channel setting, does this mean the memory path is disabled (i.e., $m_t = f_t$ as a direct bypass), or is a different configuration used? In the main model with 384 channels, how many channels do $m_t$ and $f_t$ have at the fusion point?

**Question 2:** What exactly is included in the reported fps (model inference, entropy coding/decoding, host–device I/O, preprocessing)? Additionally, do the fps numbers reflect processing a single video sequence at a time, or multiple sequences in parallel?

---

> ### Author Response · Authors · 2025-11-17
> **Rebuttal to Reviewer 94Vq, Part1**
>
> Thanks for your careful and professional review. We address your questions below.
>
> **W1.1: Reference work [R1] missed.**
>
> Thank you for the suggestion. [R1] is a foundational work demonstrating that integer neural image codecs can achieve cross-platform, bit-exact coding, forming the basis for subsequent research on integer neural image codecs. We will include reference [R1] and provide a detailed discussion in the revision.
>
> Here we briefly discuss the key difference between image and video codecs on cross-platform consistency. For image codecs, consistency can be ensured by implementing only the entropy model in integers rather than the full decoder, because round-off errors primarily impact entropy coding. In contrast, entropy coding in video codecs depends on previously decoded results, making fully deterministic decoding essential. Consequently, cross-platform image codecs can closely match the performance of floating-point models, whereas it is substantially more challenging cross-platform NVCs. To the best of our knowledge, our NVC is the first integer NVC to demonstrate near–state-of-the-art performance.
>
> **W1.2: Comparison with non-DCVC NVCs.**
>
> Thank you for the suggestion. We will include more recent SOTA methods in the revision. In our paper, we focus on all-frame coding with intra-period = –1 in YUV420, as YUV is the most widely used format in real-world applications. However, most recent SOTA NVCs differ in settings and do not release their test code or pretrained weights, making direct comparisons challenging. For instance, ConFRE [1] and DCMVC [2] test only 96 frames in RGB, and methods like GIViC [3] and L-LBVC [4] target random access scenarios. After investigating these methods, we identify two codecs suitable for direct comparison.
>
> - Among low-delay methods, only EHVC [5] releases the testing codes and pretrained weights. We test EHVC, our NVC and anchor DCVC-models at the settings (RGB BT.709 converted, all frames with intra-period -1). Our floating-point NVC significantly surpasses EHVC in compression ratio while running at least 20× faster.
> |Tested in RGB (BT.709)|UVG|MCL-JCV|HEVC B|HEVC C|HEVC D|HEVC E|Average|kMACs/pixel|Enc. / Dec. Speed (A100, 1080p)|
> |-|-|-|-|-|-|-|-|-|-|
> |DCVC-RT (fp16, CVPR 2025)|0.0|0.0|0.0|0.0|0.0|0.0|0.0|195.8|125.2 / 112.8 fps|
> |DCVC-FM (fp16, CVPR 2024)|6.1|5.1|-0.8|-14.0|-17.2|-2.4|-3.9|1073|5.0 / 5.9 fps|
> |EHVC (ACM MM 2025)|-1.8|-7.0|-9.9|-15.4|-19.8|29.6|-4.1|1311|Reported slower than DCVC-FM|
> |**Our NVC (fp16)**|-16.5|-16.5|-16.3|-14.5|-14.7|-25.4|-16.5|204.3|141.1 / 131.0 fps|
> |**Our NVC (cross-platform)**|3.0|6.6|3.7|1.8|2.8|-6.1|2.0|204.3|153.0 / 137.3 fps|
> - Very recently (2025.11.04), UI$^2$C [6] reports BD-Rate results in YUV420 using DCVC-RT as anchor. As shown in the table below, our floating-point NVC outperforms it in both speed and BD-Rate.
> |Tested in YUV420|UVG|MCL-JCV|HEVC B|HEVC C|HEVC D|HEVC E|Average|kMACs/pixel|Enc. / Dec. Speed (A100, 1080p)|
> |-|-|-|-|-|-|-|-|-|-|
> |DCVC-RT (fp16, CVPR 2025)|0.0|0.0|0.0|0.0|0.0|0.0|0.0|195.8|125.2 / 112.8 fps|
> |DCVC-FM (fp16, CVPR 2024)|6.9|6.0|0.1|-13.1|-15.7|-0.4|-2.7|1073|5.0 / 5.9 fps|
> |UI²C (arXiv 2025.11.04)|-6.1|0.9|-9.8|-16.4|-23.5|-17.7|-12.1|>233|Reported comparable to DCVC-RT|
> |**Our NVC (fp16)**|-15.9|-11.2|-16.2|-14.7|-14.8|-25.1|-16.3|204.3|141.1 / 131.0 fps|
> |**Our NVC (cross-platform)**|2.1|6.8|3.7|2.0|2.8|-6.0|1.9|204.3|153.0 / 137.3 fps|
>
>
> [1] Wu Y, Lin C, Wang Y, et al. Neural Video Compression with In-Loop Contextual Filtering and Out-of-Loop Reconstruction Enhancement. ACM MM 2025.
>
> [2] Tang C, Li Z, Bian Y, et al. Neural Video Compression with Context Modulation. CVPR 2025.
>
> [3] Gao G, Teng S, Peng T, et al. Givic: Generative implicit video compression. ICCV 2025.
>
> [4] Zhai Y, Tang L, Jiang W, et al. L-LBVC: Long-Term Motion Estimation and Prediction for Learned Bi-Directional Video Compression. DCC 2025.
>
> [5] Liao J, Wu Y, Lin C, et al. EHVC: Efficient Hierarchical Reference and Quality Structure for Neural Video Coding. Proceedings of the 33rd ACM International Conference on Multimedia.
>
> [6] Xiang H, Bian Y, Li L, et al. Real-Time Neural Video Compression with Unified Intra and Inter Coding. arXiv preprint arXiv:2510.14431, 2025.
>
> **W2: Report Computational cost.**
>
> We provide a computational complexity comparison in the table below and will include a more detailed analysis in the revised manuscript.
> |Method|Complexity|Parameters|Enc. / Dec. Speed (A100, 1080p)|
> |-|-|-|-|
> |DCVC-RT (fp16)|195.8 kMACs/pixel|20.7 M|125.2 / 112.8 fps|
> |Our NVC (fp16)|204.3 kMACs/pixel|23.2 M|141.1 / 131.0 fps|
> |DCVC-RT (int16)|56.4 MBOPs/pixel|20.7 M|28.3 / 20.9 fps|
> |Our NVC (int8)|19.6 MBOPs/pixel|23.2 M|153.0 / 137.3 fps|
>
> For the memorized temporal module, it accounts for 61.0 kMACs/pixel (about 30% of the total 204.3 kMACs/pixel). Future work will aim to further reduce the complexity of the full model, including the memory module, to better support deployment on resource-constrained devices.

---

> > ### Author Response · Authors · 2025-11-17
> > **Rebuttal to Reviewer 94Vq, Part2**
> >
> > **W3: Study on the quantization precision.**
> >
> > Thanks for the suggestion. We provide the BD-Rate comparison for different integer bit widths in the table below, showing that higher bit widths yield better coding performance. These results will be included in the revision.
> >
> > |BitWidth|Int4|Int6|Int8|Int12|Int16|
> > |-|-|-|-|-|-|
> > |BD-Rate|437.4%|4.4%|0%|-6.5%|-9.7%|
> >
> > Among the tested precisions, most GPUs (e.g., A100) support only Int4 and Int8 for Tensor Core–accelerated computation. Int4 results in excessive compression loss exceeding 400\%, so we currently adopt Int8. Exploration of lower-precision formats is left for future work.
> >
> > **W4: "Significant engineering" of CPU implementation.**
> >
> > We agree that implementing our NVC on CPU is not inherently difficult. Using PyTorch, we can perform integer computations on CPU, and **results confirm that cross-platform consistency is maintained on (1) different CPU devices and (2) across CPU and GPU devices**.
> >
> > In the original submission, our comment about the "Significant engineering" of CPU implementation primarily referred to high-performance CPU inference. The current Python-based implementation is slow, suitable only for research purposes rather than practical deployment. Achieving performance comparable to GPU inference on CPUs requires extensive kernel optimization, I/O optimization, and parallelization, hence the “significant engineering” remark. We will clarify this distinction in the revision.
> >
> > **W5: UHD comparisons**
> >
> > Thanks for the suggestion. We will revise the paper to avoid overstatement of testing cost of VTM. For UHD comparisons, we evaluate NVEnc / HEVC and VVenC / VVC as baselines. Results on Class A1 and A2 are shown below. Our NVC (int8) demonstrates substantial BD-Rate improvements over these accelerated baselines.
> >
> > |Method|Class A1|Class A2|
> > |-|-|-|
> > |NVEnc/HEVC|156.0%|132.7%|
> > |VVenC/VVC|0%|0%|
> > |**Our NVC (int8)**|-21.4%|-13.0%|
> >
> > **Minor Weakness.**
> >
> > Thank you for your careful suggestions. We will correct the typos, update the references, and provide a discussion of Figure 2 in the revised version.
> >
> > **Q1: About channel in memory module.**
> >
> > In the memory module, $m_t$ has 384 channels and $f_t$ has 256 channels. They are fused into 384 channels before being processed by the subsequent convolutional blocks. In the 0-channel memory setting, the memory path $m_t$ is bypassed entirely, and only the previous decoded feature $f_t$ is used for the following convolutional blocks.
> >
> > **Q2: About reported fps.**
> >
> > The reported fps includes model inference and entropy coding/decoding, but excludes I/O time such as loading YUV data from disk or writing decoded frames back to disk. Specifically, encoding time is measured from when the input frame is ready in GPU memory to when the output bitstream is available in CPU memory, and decoding time is measured from when the bitstream is ready in CPU memory to when the reconstructed frame is available in GPU memory. The fps numbers correspond to processing a single video sequence at a time and do not reflect parallel processing of multiple sequences.

---

> > > ### Author Response · Authors · 2025-11-27
> > >
> > > Dear Reviewer 94Vq,
> > >
> > > Thank you again for the time and effort you have dedicated to reviewing our submission. We hope our rebuttal has addressed your concerns. We would greatly appreciate any further feedback you may have, so that we can promptly clarify remaining issues if needed. Your continued engagement is invaluable in helping refine the work and clarify the concerns.
> > >
> > > Best,
> > >
> > > Authors of Submission 1445

---

### Official Review · Reviewer_s31y · 2025-11-01

**Soundness:** 4
**Presentation:** 3
**Contribution:** 2
**Rating:** 4
**Confidence:** 4

**Summary:**

This paper proposes a novel integer-centric design philosophy for neural video compression (NVC), challenging the prevailing "floating-point-centric" paradigm that first trains a full-precision model and then quantizes it into an integer implementation. The core contribution is a one-stage integer training framework that directly optimizes the integer model from scratch, aligning the training process with the target integer hardware. Extensive experiments demonstrate that the proposed integer-centric NVC achieves significant improvements in compression efficiency (measured by BD-Rate) and coding speed compared to existing cross-platform integer NVCs and even outperforms some floating-point models.

**Strengths:**

(1)	The paper effectively identifies a critical limitation in current NVC research,the performance gap introduced by quantizing floating-point models, which has a clear and compelling motivation.

(2)	The one-stage integer training framework and the Clipping Gradient Pass (CGP) mechanism are innovative solutions that directly address the challenges of training low-precision models.  The focus on hardware-friendly operations enhances the practicality and deployability of the method.

(3)   The paper provides valuable details on the integer implementation, which aids in understanding and potential reproduction of the work.

**Weaknesses:**

(1) While the overall results are impressive, the paper lacks a detailed ablation study to quantify the individual contribution of each proposed component (e.g., the impact of CGP alone, the benefit of one-stage training vs. a modified QAT, the effect of the specific integer bit-widths chosen).  This makes it difficult to assess which aspects of the framework are most critical to its success.

(2) The paper positions its work as enabling high-efficiency cross-platform NVC.  However, the experiments focus on performance metrics without explicitly demonstrating consistency or portability across different hardware platforms (e.g., CPU, GPU, mobile, FPGA).  More discussion or evidence on how the integer-centric design specifically facilitates cross-platform deployment would be beneficial.

(3) The encoding and decoding speeds of HEVC and VVC are very slow. I think the author should compare the methods with the faster x.265 and VVEnc. Moreover, as mentioned in (2), the devices used for the two types of methods (traditional or learning-based) are different (CPU vs GPU), which leads to an unfair comparison. I hope to see the performance of this method on different platforms to confirm whether this paper is really of great help in promoting the ease of use and real-time performance of learning-based video coding. If the author can solve the above problems in the subsequent rebuttal stage, I am willing to increase my score.

**Questions:**

Please see Weaknesses.

---

> ### Author Response · Authors · 2025-11-17
> **Rebuttal to Reviewer s31y**
>
> Thanks for your careful reviewing. We provide clarifications on your questions as following.
>
> **W1: Ablation on individual components.**
>
> Thank you for the suggestion. Some ablation studies on individual components are already included in the paper, and we will clarify these in the revision:
> - **Clipping Gradient Pass (CGP)**: Section 3.3 (line 252) shows that removing CGP results in a 14\% performance loss.
> - **One-Stage Training vs. Modified QAT**: Table 4 compares these on the same DCVC-RT baseline ($C_3$ and $B_3$), where one-stage training achieves about 30\% gain. On our NVC, one-stage training brings a similar 32\% improvement.
> - **Integer Bit-Widths**: The table below shows that higher bit widths provide better performance. Among the tested precisions, most GPUs (e.g., A100) support only Int4 and Int8 for Tensor Core–accelerated computation. Int4 results in excessive compression loss exceeding 400\%, so we currently adopt Int8. Exploration of lower-precision formats is left for future work.
>
> |BitWidth|Int4|Int6|Int8|Int12|Int16|
> |-|-|-|-|-|-|
> |BD-Rate|437.4%|4.4%|0%|-6.5%|-9.7%|
>
> **W2: Discussion on cross-platform consistency.**
>
> Deployment experience with traditional codecs verifies that cross-platform consistency can be **strictly ensured** through integer decoding. Theoretically, as long as deterministic integer operations are performed, consistency is guaranteed. In practice, the only potential issue is differences in overflow behavior across platforms. As discussed in Section 4.4 (line 345), our implementation theoretically avoids this issue. Therefore, our integer NVC can, in principle, ensure consistent coding across any devices that support integer computation.
>
> We validate this on **GPUs and Intel integrated graphics**, as shown in Table 2 (line 360). Following the reviewers’ suggestion, we also verify consistency on CPUs, with results **confirming cross-device reproducibility between CPU and GPU devices**. We will include these analyses and discussions in the revised manuscript.
>
> **W3: Speeds of traditional codecs are slow.**
>
> Thanks for your valuable suggestion. We test suggested traditional codecs in the below table.
>
> x.265 achieves faster decoding but suffers from a very large performance drop. Although VVenC (preset medium) is accelerated, it remains slow encoding (2 fps) and experiences over 30\% performance loss in the LDB setting. This demonstrates that traditional codecs can achieve higher speed but at the cost of substantial compression efficiency loss.
>
> NVCs are typically tested on GPUs, whereas traditional codecs are not. To enable a fair comparison, we also include NVEnc, which runs on GPUs. NVEnc achieves very high speed but at the cost of 169\% performance loss. It is worth noting that NVEnc requires chip-level hardware optimization, whereas our NVC generalizes across standard GPUs.
>
> |Method|Device|Enc./Dec(1080p)|BD-Rate|
> |---|---|---|---|
> |x265/HEVC|CPU|36.76/210.2fps|178%|
> |NVEnc&NVDec/HEVC|RTX4090|409.0/360.5fps|169%|
> |VVenC&VVdeC/VVC|CPU|2.1/118.1fps|32%|
> |OurNVC|RTX4090|145.5/124.0fps|-20.0%|

---

> ### Comment · Reviewer_s31y · 2025-11-27
>
> I think the authors have answered my doubts and I am willing to raise the score to 6.

---

> > ### Author Response · Authors · 2025-11-27
> >
> > Thank you very much for considering raising your score. We sincerely appreciate your constructive feedback throughout the review process, and we will incorporate your suggestions into the revision.

---

### Official Review · Reviewer_2sVU · 2025-11-01

**Soundness:** 2
**Presentation:** 3
**Contribution:** 2
**Rating:** 2
**Confidence:** 5

**Summary:**

This paper presents a neural video codec (NVC) that achieves both cross-platform compatibility and fast coding speed. To enable cross-platform deployment, the authors propose quantizing the NVC. Instead of following the conventional approach of converting a pretrained floating-point NVC into an integer model, they introduce a novel One-Stage Integer Training framework, which trains the integer model from scratch, that is, performing quantization-aware training (QAT) from the very beginning. To facilitate this process, they design a multiply-twice integerization method to stabilize training and propose a recurrent memorize modeling strategy to further enhance performance. Experimental results show that the proposed NVC achieves comparable rate-distortion performance to DCVC-FM, DCVC-RT, and ECM, while significantly outperforming them in terms of coding speed.

**Strengths:**

- The paper proposes an integer-centric training pipeline that enables training an integer NVC from scratch, which differs from prior works that quantize pretrained floating-point models. This design avoids potential mismatch between floating-point and integer representations and demonstrates that quantization-aware training can be effectively applied from the beginning of training.
- The multiply-twice integerization with clipped gradient pass is a key contribution that improves training stability. This approach addresses training stability and resolves the issue of gradients becoming zero in certain situations.
- The proposed temporal modeling strategy effectively captures historical context for video compression. By incorporating a recurrent memory mechanism, the method maintains temporal consistency and enhances inter-frame feature utilization, contributing to the overall compression performance.

**Weaknesses:**

- The proposed technique, such as multiply-twice integerization, lacks a strong theoretical foundation and does not guarantee improved coding performance.
- The primary performance gain appears to come from the memory model, which is simply a standard LSTM architecture. The quantization method reduces the performance significantly.
- Performance in high-bitrate scenarios is relatively weak, which may reduce the method’s practical value in applications where visual quality is prioritized. The degradation at higher bitrates suggests that the model’s capacity or quantization precision might limit its ability to preserve fine details.
- Complexity metrics (e.g., kMACs, model size) are not reported. Without these measurements, it is difficult to evaluate whether the speedup comes from architectural efficiency or simply a smaller or shallower model, making the performance claims less transparent.
- The rationale for choosing a channel size of 384 for the memory buffer is unclear. The paper only mentions that this value “balances the rate-distortion-complexity trade-off,” but it never specifies what type of complexity is being considered. It remains uncertain whether this parameter affects the overall coding speed, model size, or computational load, making it difficult to interpret the claimed balance or reproduce the design decision.
- The source of the speed advantage over DCVC-RT (int16) is ambiguous. It remains uncertain whether the improvement arises from the 8-bit integer operations, the network architecture, or implementation details. A fair comparison with a 16-bit integer version trained under the same pipeline would help isolate the key factor.
- Table 4 shows that the multiply-twice strategy does not always help when quantizing a floating-point model. This inconsistency suggests that the proposed integerization method might be sensitive to specific training conditions, limiting its general applicability.
- Comparing Table 3 and Table 6, the quantization still introduces around a 10% BD-rate increase while providing only a modest speed gain. This indicates that the proposed approach involves a nontrivial performance and complexity trade-off, rather than maintaining the same rate–distortion efficiency as the floating-point models.

**Questions:**

- Will the authors release the training scripts? Considering that the main contribution of this paper is the one-stage integer training strategy, providing the complete training pipeline would be highly beneficial for reproducibility and for the research community to further build upon this work.
- The hyperparameters K_{a, b, c, d} are empirically set, but the paper does not describe how they are determined. It would be helpful to know whether there is a systematic or data-driven way to choose these values, or if they are tuned specifically for this implementation.
- Figure 5 is somewhat unclear. When the memory channel size is set to 0, is there still any temporal information propagated to the next frame? According to Figure 3, if m_t's channel dimension is 0, there should be no temporal modeling at all, so further clarification would be useful.
- Since s_i​ and s_c​ are decoupled step sizes, is it possible for either of them to take negative values during training? If so, would a negative step size be meaningful or cause instability in the quantization process? It would be helpful if the authors could clarify whether any constraint or mechanism is applied to ensure that both step sizes remain positive.
- The paper introduces QP-aware quantization step sizes for variable bitrate. Does this imply that each QP value corresponds to a distinct set of s_i​ and s_c parameters throughout the model?
- In Table 3, for the traditional codec, is it running on the CPU, or is GPU acceleration available? If only the CPU is used, explicitly stating this in the paper would help clarify and better compare the coding speed differences among all methods.

---

> ### Author Response · Authors · 2025-11-17
> **Rebuttal to Reviewer 2sVU, Part1**
>
> Thanks for your reviewing. We provide clarifications as following.
>
> **W1: Theoretical foundation of proposed techniques.**
>
> While theoretical guarantees are valuable, they are notoriously difficult to establish for practical codec quantization. Many widely used methods rely on empirical evidence rather than formal theory, including foundational integer image codec [1] and video codec [2]. Our approach follows this practice. As shown in Table 4, removing multiply-twice integerization leads to more than a 500% BD-Rate increase, confirming that it is essential for stable one-stage training.
>
> We further provide a convergence analysis to support this claim. LSQ-style training is unstable because it may divide by extremely small step sizes when training from scratch, producing unstable gradients and rapidly growing scaling factors. Given LSQ: $\hat{v} = \left\lfloor \text{clip}(v/s, -Q_N, Q_P)\right\rceil\times s$ and multiply-twice: $\hat{v} = \left\lfloor \text{clip}(v\times s_{i}, -Q_N, Q_P)\right\rceil\times s_{c}$, we measure their BD-Rate and maximum scaling value (i.e., $\max(1/|s|, |s|)$ for LSQ and $\max(|s_i|, |s_c|)$ for ours) across training epochs for three trials. As shown in the tables, LSQ yields scaling values as large as $10^4$–$10^6$, frequently causing divergence, whereas our method consistently keeps scaling factors within a narrow and stable range of 2–4. This stability directly translates to reliable convergence and strong BD-Rate performance.
>
> | BD-Rate (%)             | epoch 10 | epoch 20 | epoch 30 | epoch 40 | epoch 50 | epoch 60 | epoch 70 | epoch 80 | epoch 90 |
> |--------------------------|----------|----------|----------|----------|----------|----------|----------|----------|----------|
> | LSQ, trial 1             | $>10^5$ | 723      | $>10^5$ | 4330     | NAN      | NAN      | NAN      | NAN      | NAN      |
> | LSQ, trial 2             | $>10^5$ | 579      | $>10^5$ | $>10^5$ | $>10^5$ | 9427     | 549      | $>10^5$ | $>10^5$ |
> | LSQ, trial 3             | $>10^5$ | 2480     | 23314    | 4171     | 650      | 543      | 2413     | $>10^5$ | $>10^5$ |
> | Multiply-twice, trial 1  | 4068     | 135      | 57       | 56       | 50       | 36       | 10       | 0.8      | 0.0      |
> | Multiply-twice, trial 2  | 3866     | 130      | 57       | 51       | 52       | 44       | 14       | 3.0      | 1.2      |
> | Multiply-twice, trial 3  | 4106     | 133      | 58       | 54       | 43       | 37       | 21       | 2.3      | 1.1      |
>
> | Max scaling value        | epoch 10 | epoch 20 | epoch 30 | epoch 40 | epoch 50 | epoch 60 | epoch 70 | epoch 80 | epoch 90 |
> |--------------------------|----------|----------|----------|----------|----------|----------|----------|----------|----------|
> | LSQ, trial 1             | $10^1$  | $10^3$  | $10^5$  | $10^4$  | NAN      | NAN      | NAN      | NAN      | NAN      |
> | LSQ, trial 2             | $10^1$  | $10^2$  | $10^6$  | $10^4$  | $10^4$  | $10^3$  | $10^5$  | $10^6$  | $10^6$  |
> | LSQ, trial 3             | $10^1$  | $10^4$  | $10^5$  | $10^4$  | $10^4$  | $10^3$  | $10^6$  | $10^4$  | $10^4$  |
> | Multiply-twice, trial 1  | 2.1      | 3.0      | 3.3      | 3.8      | 4.1      | 4.1      | 4.1      | 4.0      | 4.0      |
> | Multiply-twice, trial 2  | 2.1      | 3.0      | 3.3      | 3.9      | 4.0      | 4.0      | 4.0      | 4.0      | 4.0      |
> | Multiply-twice, trial 3  | 2.0      | 3.0      | 3.3      | 3.9      | 4.0      | 4.1      | 4.1      | 4.1      | 4.1      |
>
> [1] Ballé, J., Johnston, N., \& Minnen, D. (2019). Integer Networks for Data Compression with Latent-Variable Models. International Conference on Learning Representations (ICLR).
>
> [2] Van Rozendaal T, Singhal T, Le H, et al. MobileNVC: Real-time 1080p neural video compression on a mobile device. Proceedings of the IEEE/CVF Winter Conference on Applications of Computer Vision.
>
> **W2 \& W8: Quantization causes a 10\% BD-rate increase.**
>
> Quantization inevitably introduces performance loss, and an Int8 NVC cannot theoretically match a floating-point model due to precision limits. The central question is thus how small this gap can be while still enabling deterministic integer decoding. In this respect, our approach achieves one of the smallest reported degradations. For example, MobileNVC shows roughly an 80\% BD-Rate increase after quantization, whereas our scheme limits this to about 10\%.
>
> Although integer decoding incurs a performance penalty, it is a fundamental requirement to guarantee deterministic outputs across platforms, as required in practical video codecs such as H.264/AVC, H.265/HEVC, and H.266/VVC. Floating-point NVCs cannot support cross-device decoding and are thus unusable in real deployments, even if they offer slightly better performance. Therefore, we view the observed 10\% BD-Rate increase not as a drawback, but as strong evidence that our quantization strategy maintains rate–distortion efficiency exceptionally well among existing integer NVC approaches.

---

> > ### Author Response · Authors · 2025-11-17
> > **Rebuttal to Reviewer 2sVU, Part2**
> >
> > **W3: High-bitrate performance drop.**
> >
> > As noted in our response to W2, all quantized neural codecs exhibit reduced performance due to limited numerical precision. The high-bitrate degradation observed here is therefore a general consequence of quantization rather than a limitation of our method. In fact, our approach mitigates this effect more effectively than prior work. In our NVC, Int8 quantization leads to only about a 0.1 dB PSNR drop at high rates. For comparison, DCVC-RT with higher-precision Int16 quantization shows a similar 0.1 dB loss, whereas MobileNVC suffers more than a 1 dB drop with Int8.
> >
> > **W4 \& W6: Report complexity metrics and analysis source of speed advantage.**
> >
> > We report the kMACs/pixel and model parameters in the table below. Our NVC has complexity metrics comparable to DCVC-RT, confirming that the speedup does not come from a smaller or shallower architecture. The improvement primarily stems from quantization, and we will include these metrics in the revised version.
> >
> > DCVC-RT [3] notes that its low Int16 speed results from limited hardware support for 16-bit integer operations. In contrast, modern accelerators provide highly optimized Int8 kernels, enabling substantial acceleration. Our comparison further isolates this effect: our Int16 model runs at a speed similar to DCVC-RT, while switching to Int8 delivers over a 5× speedup. This shows that Int8 precision is the dominant source of the performance gain.
> >
> > |Method|kMACs/pixel|Parameters|Enc./Dec.Speed(A100,1080p)|
> > |-|-|-|-|
> > |DCVC-RT(int16)|195.8|20.7M|28.3 / 20.9fps|
> > |OurNVC(int16)|204.3|23.2M|32.3 / 24.2fps|
> > |OurNVC(int8)|204.3|23.2M|153.0 / 137.3fps|
> >
> > [3] Jia Z, Li B, Li J, et al. Towards practical real-time neural video compression. Proceedings of the Computer Vision and Pattern Recognition Conference.
> >
> > **W5: Choice of memory module channel size.**
> >
> > In Figure 5, the channel size determines both the memory buffer width and the convolutional network width within the memory module. A larger buffer requires a larger module to process the stored information effectively. The table below shows how computational complexity varies with channel size. We selected 384 because it matches the overall complexity of DCVC-RT (195.8 kMACs/pixel), providing a balanced trade-off between coding performance and computational load.
> >
> > |ChannelSize|0|128|256|384|512|
> > |-|-|-|-|-|-|
> > |kMACs/pixel|143.3|150.8|170.1|204.3|250.2|
> > |Parameters|15.7M|16.6M|19.0M|23.2M|28.9M|
> >
> > **W7: Multiply-twice strategy does not help finetuning.**
> >
> > We do not claim that the multiply-twice strategy is effective when used in isolation. Rather, we emphasize that its benefit emerges in combination with one-stage integer training. This combination is essential: one-stage training provides the flexibility needed to recover performance, while the multiply-twice strategy stabilizes the optimization process and prevents divergence. We believe that this synergistic approach has broad applicability across diverse training scenarios.
> >
> > **Q1: Release of training scripts.**
> >
> > We will release the training scripts upon acceptance.
> >
> > **Q2: Choosing Hyperparameters $K_{a, b, c, d}$**
> >
> > We determine these hyperparameters based on the numerical ranges of the values they scale. For example, as noted in line 834, the corresponding floating-point values fall within $[-32.0, 31.999]$, so we set $K_a=2^{10}=1024$ to map it within Int16 range $[-32768,32767]$. This minimizes quantization error while avoiding overflow. The same principle is applied when choosing the other hyperparameters. This procedure is a straightforward engineering choice rather than a data-driven optimization, and the values are selected empirically according to the ranges they must represent.
> >
> > **Q3: Illustration of memory channel size.**
> >
> > In Figure 5, encoding the next frame $I_{t+1}$ uses both the current decoded feature $f_t$ and the history memory $m_{t-1}$ as conditions. When the memory channel size is set to 0, $m_{t-1}$ is empty, so only $f_t$ is used to encode $I_{t+1}$, analogous to the temporal context used in DCVC-RT. We will clarify this in the revision.

---

> > > ### Author Response · Authors · 2025-11-17
> > > **Rebuttal to Reviewer 2sVU, Part3**
> > >
> > > **Q4: Negative values for decoupled step sizes.**
> > >
> > > Step sizes in our NVC can be negative, and in fact, negative values naturally occur. In one-stage integer training, step sizes are not learned to “quantize” a value but to “scale” it to the correct magnitude, so negative value is also meaningful for scaling. For instance, in Equation 4 ($\hat{v} = \left\lfloor \text{clip}(v\times s_{i}, -Q_N, Q_P)\right\rceil\times s_{c}$), using $(s_i, s_c)$ produces almost the same result as $(-s_i, -s_c)$, meaning the sign of the step size alone is largely irrelevant.
> > >
> > > Negative scales do not cause instability. For example, updating $s_i$ by $\mathrm{d}s_i$ changes $v \cdot s$ by $v \cdot \mathrm{d}s$, which is independent of the sign of $s_i$. If $v=1000, s_i=0.001$, an update of $\mathrm{d}s_i=0.002$ changes the output $1$ to $3$, while $\mathrm{d}s=-0.002$ changes the output $1$ to $-1$. In both cases, the magnitude of change is the same (i.e., change by 2), indicating a similar impact on subsequent calculations.
> > >
> > > **Q5: QP-aware quantization step sizes.**
> > >
> > > In our NVC, each QP value corresponds to a distinct set of quantization step sizes. Since each step size is a single real value, this introduces only a negligible increase in the total number of parameters.
> > >
> > > **Q6: Test device of traditional codec.**
> > >
> > > In Table 3, the traditional codec is evaluated on high-end CPU devices, as GPU implementations are not available. To further contextualize speed comparisons, we also evaluate chip-level optimized NVEnc/NVDec on the same RTX 4090 GPU, as shown below. While NVEnc achieves very high speed, this comes at the cost of substantial performance loss. We will clarify this in the revision.
> > >
> > > |Method|Device|Enc. / Dec(1080p)|BD-Rate(VTM)|
> > > |---|---|---|---|
> > > |NVEnc & NVDec / HEVC|RTX4090|409.0 / 360.5fps|169%|
> > > |OurNVC|RTX4090|145.5 / 124.0fps|-20.0%|

---

> > > > ### Author Response · Authors · 2025-11-27
> > > >
> > > > Dear Reviewer 2sVU,
> > > >
> > > > Thank you again for the time and effort you have dedicated to reviewing our submission. We hope our rebuttal has addressed your concerns. We would greatly appreciate any further feedback you may have, so that we can promptly clarify remaining issues if needed. Your continued engagement is invaluable in helping refine the work and clarify the concerns.
> > > >
> > > > Best,
> > > >
> > > > Authors of Submission 1445

---

### Author Response · Authors · 2025-11-25
**Follow-up on Rebuttal Discussion**

Dear Reviewers,

Thank you again for the time and effort you have devoted to reviewing our submission. We would like to kindly follow up on the rebuttal discussion. It has been over a week since we submitted our response, and the discussion period is entering its final week. If there are any remaining questions or points that would benefit from further clarification, we would be very glad to address them promptly.

Warm regards,
Authors of Submission 1445

---

### Author Response · Authors · 2025-12-01
**Summary of Discussion Status and Rebuttal Highlights**

Dear AC,

Given the incident during the discussion period, we would like to provide a brief summary of the review status and rebuttal outcomes for your reference.

****
**1. Initial Review**

We received very positive feedback from Reviewers **dbJf** and **RwNV** (Score: 6, Confidence: 5). Reviewer **s31y** (Score: 4) expressed a **willingness to raise their score** if specific issues were resolved.

****
**2. Discussion stage**

We responded to all reviewers within **3 days**. After that:
- Reviewer **s31y** confirmed the doubts were addressed and raised the score from 4 to 6 (before the incident)
- Reviewer **RwNV** engaged in further constructive discussion. We believe this discussion could have led to a positive outcome if not for the incident.
- Unfortunately, despite our detailed rebuttal and two reminders, reviewer **94Vq** and **2sVU** have not responded in the **11 days**. It is very regretful that no discussion has taken place during such a long window, even under ICLR’s open discussion policy.

****
**3. Rebuttal Content**

For your convenience, we categorize the reviewers’ concerns into three groups.

**3.1. Shared by multiple reviewers, claimed to be addressed by responsed reviewer**

Although the reviewer 94Vq and 2sVU did not respond, their primary concerns overlapped with those of the active reviewers. Since the active reviewers confirmed these issues were resolved, we believe the concerns of the unresponsive reviewers are also effectively addressed.

| Concern | Shared By | Status |
| :--- | :--- | :--- |
| Theoretical foundation of multiply-twice | 2sVU, dbJf, RwNV | **Resolved** (Confirmed by RwNV) |
| Reporting complexity metric | 2sVU, 94Vq, dbJf, RwNV | **Resolved** (Confirmed by RwNV) |
| Memory channel size and designs | 2sVU, 94Vq, dbJf, RwNV | **Resolved** (Confirmed by RwNV) |
| Ablation study on bit widths | s31y, 94Vq, dbJf | **Resolved** (Confirmed by s31y) |
| Cross-device consistency on CPU | s31y, 94Vq, dbJf | **Resolved** (Confirmed by s31y) |
| Testing faster traditional codecs | 2sVU, s31y | **Resolved** (Confirmed by s31y) |


**3.2. Requests for additional experimental results (not actual weaknesses)**

Most unanswered comments focus on requests for additional results **rather than weaknesses** of the work. For example, comparisons with non-DCVC codecs, testing an intra-period of 32, or adding more ablations.

We have provided all of these results in the rebuttal, and the new findings are fully consistent with our original claims. We believe these points would have received positive feedback had the discussion period continued.


**3.3. Claimed weakness with no reviewer response**

Only one concern falls under this category and may require further clarification. Reviewer **2sVU** argues that **our NVC exhibits a 10% BD-rate increase compared to the floating-point model**. We do not consider this a true weakness. Our rebuttal stated:

> Quantization inevitably introduces performance loss, and an Int8 NVC cannot theoretically match a floating-point model due to precision limits. The central question is thus how small this gap can be while still enabling deterministic integer decoding. In this respect, our approach achieves one of the smallest reported degradations. For example, MobileNVC shows roughly an 80% BD-Rate increase after quantization, whereas our scheme limits this to about 10%.
>
> Although integer decoding incurs a performance penalty, it is a fundamental requirement to guarantee deterministic outputs across platforms, as required in practical video codecs such as H.264/AVC, H.265/HEVC, and H.266/VVC. Floating-point NVCs cannot support cross-device decoding and are thus unusable in real deployments, even if they offer slightly better performance. Therefore, we view the observed 10% BD-Rate increase not as a drawback, but as strong evidence that our quantization strategy maintains rate–distortion efficiency exceptionally well among existing integer NVC approaches.

As stated, integer NVC is a fundamental requirement, and it is expected to underperform its floating-point counterpart. Our goal is to develop a strong integer NVC, not to surpass a floating-point model, which is inherently impossible due to precision limitations.

---

### Meta-Review · Area_Chair_DyVB · 2026-01-02

**Summary:**

The quantization method reduces the performance significantly (2sVU), especially for higher bitrate (2sVU, dbJf).
Very closely related prior works are not cited, discussed and compared (94Vq). By a literature tracking based on [1] which has 94 citations, there exist a series of previous works dedicated for cross-platform data coding regarding images, videos and graphs [1-9]. Quantizing only the entropy part or the whole decoder does not change the nature of the quantization problem, as modern general model quantization are commonly applied to very deep neural networks [10], and can be adapted for integer neural codec [1-9].



Without discussion and comparison with some of those previous works regarding efficiency and RD performance, it is impossible to position the contribution of this manuscript.
Cross-platform codec based on interger networks are closely related to the popular model quantization field, to justify a novel contribution, it is necessary to verify main-stream model quantization techniques and previous cross-platform nerual codec are not sufficient.
Besides, all the reviewers raise many detailed issues regarding motivation, ablation study, and clarity issues,  suggesting the manuscript requires substantial improvement.

To sum up, as pointed out by some reviewers, it is known that integer network can be used for cross-platform codec [1-9]. However, quantization leads to the tradeoff between parameter precision and model performance. To justify the novelty and performance of the proposed approach, it is important to show the proposed method achieves better tradeoff compared with previous cross-platform works and easily accessible main-stream model quantization techniques [10]. LSQ 2020 compared in the manuscript is relatively old method for general model quantization.






[1] Ballé, J., Johnston, N., & Minnen, D. (2019). Integer Networks for Data Compression with Latent-Variable Models. International Conference on Learning Representations (ICLR).
[2] Tian, Kuan, et al. "Effortless cross-platform video codec: A codebook-based method." arXiv preprint arXiv:2310.10292 (2023).
[3] Koyuncu, Esin, et al. "Device interoperability for learned image compression with weights and activations quantization." 2022 Picture Coding Symposium (PCS). IEEE, 2022.
[4] He, Dailan, et al. "Post-training quantization for cross-platform learned image compression." arXiv preprint arXiv:2202.07513 (2022).
[5] Shi, Junqi, Ming Lu, and Zhan Ma. "Rate-distortion optimized post-training quantization for learned image compression." IEEE Transactions on Circuits and Systems for Video Technology 34.5 (2023): 3082-3095.
[6] Tian, Kuan, et al. "Towards real-time neural video codec for cross-platform application using calibration information." Proceedings of the 31st ACM International Conference on Multimedia. 2023.
[7] Zhang, Ge, et al. "Integer Network for Cross Platform Graph Data Lossless Compression." 2022 IEEE International Conference on Multimedia and Expo Workshops (ICMEW). IEEE, 2022.
[8] Koyuncu, Esin, et al. "Quantized decoder in learned image compression for deterministic reconstruction." ICASSP 2024-2024 IEEE International Conference on Acoustics, Speech and Signal Processing (ICASSP). IEEE, 2024.
[9] Conceição, Ruhan, et al. "Cross-Platform Neural Video Coding: A Case Study." 2025 IEEE International Symposium on Circuits and Systems (ISCAS). IEEE, 2025.
[10] Liu, Kai, et al. "Low-bit model quantization for deep neural networks: A survey." arXiv preprint arXiv:2505.05530 (2025).

**Reviewer Concerns:**

Solved:
Complexity metrics (e.g., kMACs, model size) are not reported.
The rationale for choosing a channel size of 384 for the memory buffer.
Lacks a detailed ablation study to quantify the individual contribution of each proposed component.
The experiments focus on performance metrics without explicitly demonstrating consistency or portability across different hardware platforms.
Comparison with non-DCVC NVCs.
How coding efficiency and complexity trade off under more aggressive (e.g., 6- or 4-bit) or less aggressive (e.g., 10-, 12-, 16-bit) precisions.
Many detailed issues regarding ablation study and clarity.


Still outstanding concerns:
The primary performance gain appears to come from the memory model, which is simply a standard LSTM architecture. The quantization method reduces the performance significantly (2sVU), especially for high bitrate (dbJf).
The source of the speed advantage over DCVC-RT (int16) is ambiguous (2sVU).
Related work coverage and comparison with previous cross-platform neural codec techniques (94Vq).
Limited theoretical justification (dbJf, 2sVU).

**Reviewer Scores:**

2sVU might keep 2 as the concern on performance is not solved during the rebuttal.
s31y might increase from 4 to 6 as he stated.
94Vq might keep 4 because the manuscript is still not properly positioned and misses discussion and comparison regarding previous cross-platform neural codec.
dbJf might keep 6.
RwNV might keep 6.

---

### Decision · Program_Chairs · 2026-01-26

Reject